# Regulation of cellular sterol homeostasis by the oxygen responsive noncoding RNA *lincNORS*

Xue Wu et al.[#]

We hereby provide the initial portrait of *lincNORS*, a spliced lincRNA generated by the MIR193BHG locus, entirely distinct from the previously described miR-193b-365a tandem. While inducible by low $O_2$ in a variety of cells and associated with hypoxia in vivo, our studies show that *lincNORS* is subject to multiple regulatory inputs, including estrogen signals. Biochemically, this lincRNA fine-tunes cellular sterol/steroid biosynthesis by repressing the expression of multiple pathway components. Mechanistically, the function of *lincNORS* requires the presence of RALY, an RNA-binding protein recently found to be implicated in cholesterol homeostasis. We also noticed the proximity between this locus and naturally occurring genetic variations highly significant for sterol/steroid-related phenotypes, in particular the age of sexual maturation. An integrative analysis of these variants provided a more formal link between these phenotypes and *lincNORS*, further strengthening the case for its biological relevance.

[#]A list of authors and their affiliations appears at the end of the paper.

Cells monitor intracellular oxygen ($O_2$) tension using specific dioxygenase enzymes that act as negative regulators for hypoxia-inducible factors (HIF) and their downstream transcriptional programs[1,2]. Although the core of this mechanism is known to operate in essentially all metazoans, decreased $O_2$ (hypoxia) also elicits context-specific responses, which remain incompletely understood. Similar to other signaling pathways, valuable insights to this end continue to emerge from noncoding transcriptome studies[3]. During the past decade, various types of non-coding RNAs have been shown to respond to hypoxia, potentially functioning as feedback elements that impact various branches of this response. Among the best documented is miR-210, the prototypical hypoxia microRNA[4] thought to fine-tune mitochondrial activity to environmental $O_2$ availability[5]. More recently, the oxygen-sensitive noncoding transcriptome has been expanded to include hypoxia-inducible lncRNAs, such as NEAT1, MALAT1, MEG3, H19, NLUCAT1, HOTAIR, HIF-1-AS2, and MIR210HG[6–11]. Although these lncRNAs exert biochemical and phenotypical effects during experimental oxygen deprivation, there is extensive evidence to support their involvement in a broader spectrum of responses[12]. Furthermore, their overall relevance in physiology and disease remains insufficiently understood.

Here we present an initial analysis of the mature lincRNA product of MIR193BHG locus, henceforth termed **N**oncoding **O**xygen-Sensitive **R**egulator of **S**terol Homeostasis (lincNORS). This locus was previously investigated exclusively in the context of the miR-193b-365a tandem. Our investigation reveals a connection between lincNORS, oxygen-sensing, and steroid hormone responses, adding to the growing evidence of the interdependence and overlap between these networks[13]. Our study identifies a previously unrecognized modulator of the cellular steroid biosynthesis pathways and points to its relevance for specific human phenotypes.

## Results

**Identification and characterization of lincNORS.** We first screened noncoding transcripts induced by decreased $O_2$ tension in MCF7 breast cancer cells. RNA-seq analysis captured the expected hypoxic coding signature and the increased abundance of NEAT1, MALAT1, MIR210HG, and HIF1A-AS2, known as non-coding hypoxic transcripts (Fig. 1a, Supplementary Fig. 1a, and Supplementary Data 1)[6–9]. Among the top upregulated noncoding RNAs, we identified a previously uncharacterized lncRNA, ENSG00000262454/RP11-65J21.3/MIR193BHG, corresponding to the genomic locus that hosts the miR193b-365a tandem (Supplementary Fig. 1b). The miR-193b and miR-365a sequences are intronic to this locus and are therefore lost during splicing, justifying a specific designation for the mature lncRNA. Our decision to focus on this transcript was due to its consistent and wide-range response to oxygen deprivation in a broad spectrum of human cell lines, both transformed and non-transformed (Fig. 1b, c and Supplementary Fig. 1c). Although early studies were performed under severe hypoxia (0.2% $O_2$), predominantly encountered in the tumor environment, subsequent studies were performed under 1–2% $O_2$, an in vitro approximation of physiologically low $O_2$[3,14]. Interestingly, no significant induction of miR-193b and miR-365a was observed upon $O_2$ deprivation, in contrast to miR-210, which exhibited the expected induction (Fig. 1d). The result reflects published microRNA-seq and RNA-seq data for MCF-7 cells, whereas these miRNAs are rarely reported as hypoxia-inducible[15–17]. A potential explanation is that miRNA biogenesis is partially impaired during hypoxia[18,19].

Analysis of The Cancer Genome Atlas (TCGA) data sets revealed a positive correlation between lincNORS expression and a panel of well-established hypoxia signature genes in several cancers, including triple-negative breast cancer (TNBC) and pancreatic adenocarcinoma (PAAD) (Fig. 1e and Supplementary Data 2). Using a hypoxia score, calculated based on the hypoxia metagenes established by Buffa et al.[20], we further showed that in both TNBC and PAAD, the in vivo lincNORS levels were more closely associated with the hypoxia state than NEAT1, a lncRNA known to respond to oxygen deprivation (Supplementary Fig. 1d).

To begin characterization of this lncRNA, we used 5′-RNA ligation-mediated rapid amplification of cDNA ends (5′-RACE) which identified two main transcription start sites (TSS), corresponding to ENST00000570945/MIR193BHG-001 and ENST00000634265/MIR193BHG-004, matching cap analysis of gene expression (CAGE) information from the FANTOM project (Supplementary Fig. 2a, b)[21]. Furthermore, the two polyadenylation (PolyA) sites identified by 3′ RACE agree with Cancer Genome PolyA Site & Usage data (Supplementary Fig. 2a, b). A closer examination of lincNORS revealed well-recognized features of a bona fide lncRNA. First, multiple coding-potential analyses produced consistently low scores (Supplementary Fig. 2c), and the SORFS mass spectrometry database (http://sorfs.org/database) did not list any peptides associated with this locus at the time of writing. Second, the locus is separated from the flanking ones. Third, as expected for a lncRNA, lincNORS is expressed in a tissue-specific pattern, with a higher abundance in hormone-responsive tissues and skeletal muscle (Supplementary Fig. 3). Furthermore, GTEx data sets indicate an overall abundance comparable to, or higher than, the well-established lncRNAs HOTAIR, CCAT1, and MIR210HG (Supplementary Fig. 3).

Cell fractionation revealed a predominant, albeit not exclusive, nuclear localization of lincNORS in MCF7 cells (Fig. 1f). Furthermore, the majority of nuclear lincNORS is chromatin-associated, a relatively common feature of lncRNAs involved in gene expression regulation. Individual lincNORS loci were visualized by hybridization with custom-designed RNAscope® probes, a pattern consistent with nuclear enrichment and hypoxic responsiveness (Supplementary Fig. 1e).

**Upstream molecular determinants of lincNORS abundance.** To gain deeper insights into the hypoxic induction of lincNORS, we profiled the response at the primary transcript level using the nascent RNA Bru-seq method (Supplementary Data 3)[22]. As shown in Fig. 2a, the MIR193BHG locus exhibited a significantly increased signal in hypoxic cells, consistent with a transcriptional effect. As internal controls for the assay, canonical hypoxia-inducible genes displayed the expected activation (Supplementary Fig. S4a).

Based on the evidence for a transcriptional response at this locus, we further tested the possible role of HIF-α isoforms in loss-of-function experiments. Although HIF-1α depletion had no measurable effect on lincNORS levels (Supplementary Fig. 4b), HIF-2α knockdown blunted the hypoxic accumulation of lincNORS in breast cancer cells (Fig. 2b). Supporting evidence for the role of HIF-2 was obtained using 786-O kidney cancer cells, which are documented to overexpress HIF-2α and corresponding targets regardless of $O_2$ tension due to pVHL E3 ligase inactivation[1]. Similarly, the sensitivity of lincNORS to low $O_2$ is compromised in VHL-deficient cells and rescued by restoring VHL expression (Fig. 2c). This result is consistent with results reported by Zhang et al. listing MIR193BHG (lincNORS) among the HIF-2 targets[23]. The preferential role of HIF-2 has

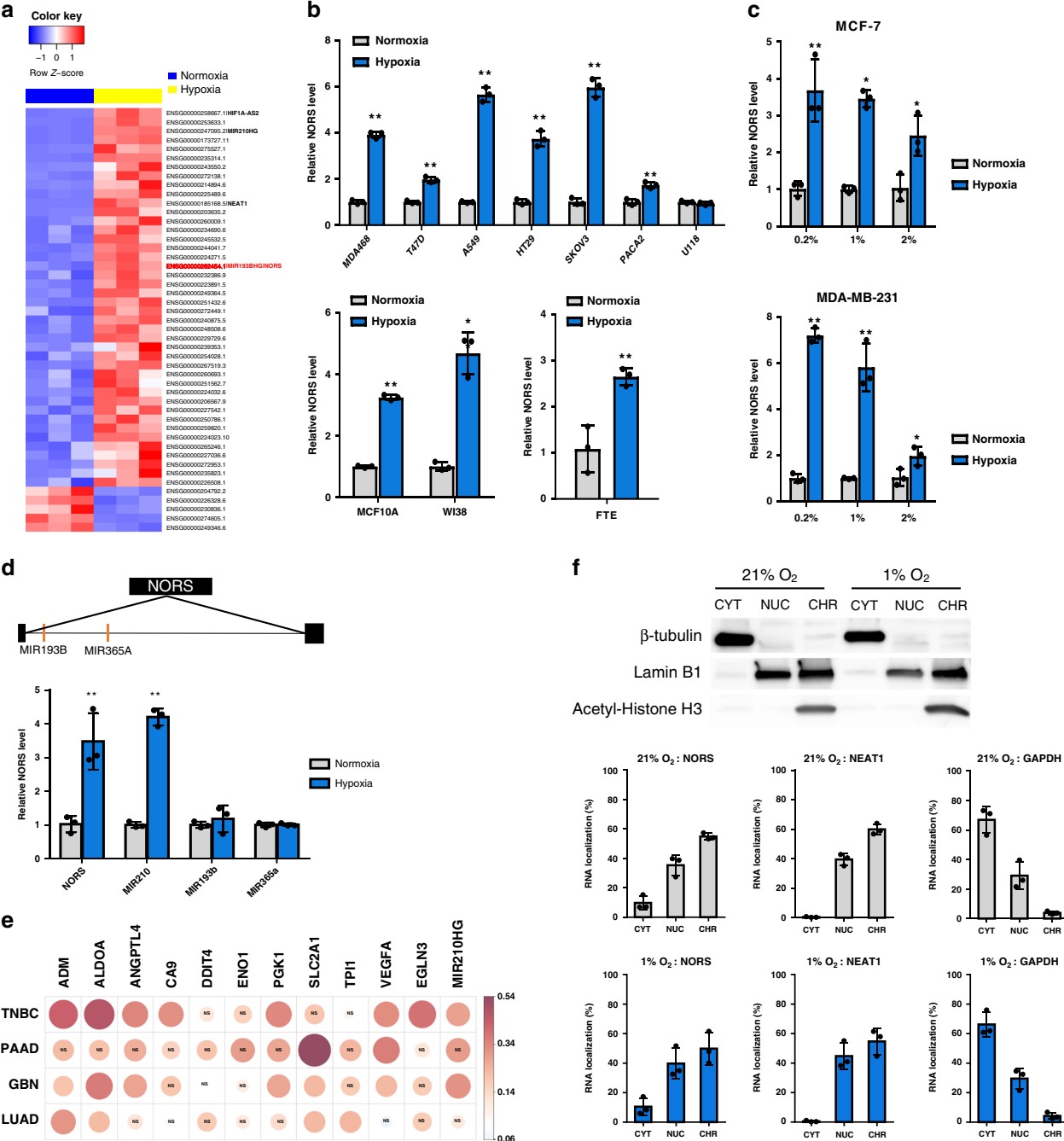

**Fig. 1 lincNORS is an oxygen-regulated, nucleus-located lncRNA. a** Heatmap depicting the expression of 47 lincRNAs significantly upregulated (fold change ≥ 2 or ≤ 0.5) in MCF-7 cells under hypoxia (1% $O_2$, 24 h) versus normoxia (21% $O_2$, 24 h). Color intensity represents RPKM scaled by the row. Gene symbol of previously reported hypoxia-inducible lincRNAs and *lincNORS* was added next to the Ensembl ID. **b** Validation of *lincNORS* induction by hypoxia across diverse cell types, including both tumor (0.2% $O_2$) and non-tumorigenic cell lines (1% $O_2$), and fallopian tube epithelial cells (2% $O_2$). *lincNORS* RNA level was measured by qPCR. All the samples were cultured in hypoxia for 24 h except FTE cells (48 h). Data represent mean ± SD from three biological replicates (\*\**p* < 0.01, \**p* < 0.05, two-sided Student's *t*-test). **c** qPCR analysis of *lincNORS* expression under 0.2%, 1%, 2% $O_2$ in MCF-7 and MDA-MB-231 cell lines. Data represent mean ± SD from three biological replicates (\*\**p* < 0.01, \**p* < 0.05, two-sided Student's *t*-test). **d** qPCR analysis of *miR193b, miR365a, miR210,* and *lincNORS* expression in MCF-7 cells cultured under 21% or 1% $O_2$. A diagram of *lincNORS* and its intronic miRNAs is shown above. *miR210* served as a positive reference for hypoxia-induced miRNA. Data represent mean ± SD from three biological replicates (\*\**p* < 0.01, two-sided Student's *t*-test). **e** Spearman's correlation between *lincNORS* and well-established hypoxia signature genes in triple-negative breast cancer (TNBC), pancreatic adenocarcinoma (PAAD), glioblastoma (GBM), and lung adenocarcinoma (LUAD) from TCGA. The size of the dots is correlated with the strength of the Spearman's coefficient. The red color represents a positive correlation while the blue color represents a negative correlation. The corresponding scale is presented next to the image. "NS" was added inside the dots where the correlation is not significant. The significance is defined as *p* < 0.05. **f** MCF-7 cells were harvested and fractionated after 24 h culture in 21% $O_2$ or 1% $O_2$. The RNA distribution of *lincNORS, NEAT1,* and *GAPDH* in cytoplasmic (CYT), nucleoplasm (NUC), and chromatin-associated (CHR) fractions were assessed by qPCR. *NEAT1* and *GAPDH* were included as nuclear and cytoplasmic controls. Data represent mean ± SD from three biological replicates. Western blot was performed to confirm cell fractions and a representative image from three independent experiments was shown.

been previously described for hypoxia lncRNAs, in particular *NEAT1*, *MALAT1*, and *NLUCAT1*[6,11].

A survey of the genomic region surrounding *lincNORS* using available ChIP-seq data sets (GSE28352) did not reveal functional HIF recruitment sites within the *lincNORS* proximal regulatory region, in contrast to standard hypoxia responders typically driven by HIF-1 from proximal promoters (e.g., CA9 and *miR210/MIR210HG*) (Fig. 2d)[24]. Based on the knowledge that HIFs (and in particular HIF-2) sometimes transactivate from distant enhancers, we identified major peaks at the neighboring locus *LINC02130* (Fig. 2e and Supplementary Fig. 5a)[24,25]. This locus is a plausible distal regulatory element based on (1) ENCODE genome segmentation features (Supplementary Fig. 5b); (2) chromatin interaction with the *lincNORS* proximal regulatory region (RNAPII ChIA-PET GSE33664) (Fig. 2e)[26]; (3) LINC02130 transcript is much less abundant than *lincNORS* in tissues and cell lines (Fig. 2e—GTEx track summary and Supplementary Fig. 1b for MCF7 cells); and (4) evidence of activation by hypoxia but the corresponding transcript has very low abundance (Supplementary Fig. 1b). Interestingly, the HIF prolyl-hydroxylase inhibitor DMOG was much less effective in inducing *lincNORS* than low $O_2$ (Supplementary Fig. S4c, d). Despite inducing the expected hypoxic responses (increased HIF-2 protein abundance; CA9 transcript upregulation), DMOG impact on *lincNORS* was modest in MDA-MB-231 and reproducibly absent in MCF7. In conclusion, while these data indicate that *lincNORS* is a hypoxia-inducible gene, they also show that it is an atypical one, prompting us to investigate additional details on *lincNORS* upregulation.

Detailed examination of the *lincNORS* genomic neighborhood revealed hormonal response hallmarks, including DNase I hypersensitivity peaks that are further stimulated by estradiol (E2) treatment (Supplementary Fig. 6a), as well as ChIP-seq peaks for estrogen receptor (ER), ER pioneer factors (FOXA1, GATA3) and ER coactivators (NCOA3/SRC3) in relevant cell types (Supplementary Fig. 6b). As is often the case for estrogen-responsive genes, the major regional ER peak resides at the putative *LINC02130* enhancer (Supplementary Fig. S6a)[27]. Accordingly, a mega trans ER enhancer was identified at this site in the recent study by Rosenfeld and colleagues (Supplementary Fig. 6c)[28]. Finally, our analysis of TCGA RNA-seq data sets detected significantly higher *lincNORS* expression in ER/progesterone receptor (PR) positive breast tumors compared to their negative counterparts (Fig. 3a). Collectively, this information prompted us to directly test the response of *lincNORS* to E2 treatment. In short-term experiments, hormonal treatment resulted in reproducible (if not marked) *lincNORS* induction (Fig. 3b). More strikingly, *lincNORS* expression was dramatically reduced in long-term estrogen deprivation (LTED) of MCF7 cells (Fig. 3c).

ChIP-seq data sets also suggest a potential role of FOSL2 in the regulation of *lincNORS*, a hypoxia-inducible AP-1 family member (Supplementary Fig. 5a)[29]. Using a loss-of-function approach, we confirmed the positive impact of FOSL2 on *lincNORS* expression in breast cancer cells (Supplementary Fig. 5c), predominantly at a lower oxygen concentration. Based on the evidence that FOSL2 responds to multiple stimuli, including steroid hormones, its role in the regulation of *lincNORS* may not be limited to oxygen-related responses[30,31].

**lincNORS coordinately modulates the sterol biosynthesis program.** To gain insight into *lincNORS* biochemical functions, we downregulated it using a combination of siRNA/shRNA, similarly to the recently published studies on the lncRNAs *CHROME* and *SCAT*[32,33] (Supplementary Fig. 7a). For transient interventions,

we first validated non-overlapping siRNAs targeting the major exon of *lincNORS* (present in all isoforms). Based on their differential ability to knockdown *lincNORS* expression, these will be henceforth referred to as siNORS_s (strong) and siNORS_w (weak) (Supplementary Fig. 7b).

We next performed RNA-seq analysis in hypoxic MCF7 cells to determine the consequences of *lincNORS* downregulation (Supplementary Fig. 8a and Supplementary Data 4). Differentially regulated transcripts were subjected to Gene Set Enrichment Analysis (GSEA) and Ingenuity Pathway Analysis (IPA), both of which identified the upregulation of cholesterol biosynthesis as the top transcriptomic signature (Fig. 4a and Supplementary Fig. 8b). As illustrated in Fig. 4b and Supplementary Fig. 8c, the program responsible for the de novo synthesis of cholesterol and related steroids was coordinately upregulated by *lincNORS* siRNA, with significant upregulation of many individual genes. A similar effect was evidenced in ER-negative MDA-MB-468 cells using the AmpliSeq platform (Fig. 4c, d and Supplementary Data 5). Individual genes were validated by qPCR in a diverse cell panel (Fig. 4e, f and Supplementary Fig. 8d). The impact of *lincNORS* may extend beyond biosynthesis, with negative effects on the *STARD4* transport regulator and potentially other steroid network components (Fig. 4g and Supplementary Fig. 8f). Similar effects (Fig. 4g) were demonstrated on *SCD* (stearoyl CoA desaturase), a gene often coregulated with a sterol biosynthesis program. Evidence that *lincNORS* was successfully repressed came from several lines of evidence: (1) The knockdown efficiency of siNORS_s was greater than that of siNORS_w and overall it exhibited a stronger impact on sterol biosynthesis transcripts; (2) Although this response is also detectable under high $O_2$, it was generally more obvious at lower oxygenation, as expected based on *lincNORS* abundance; (3) *lincNORS* downregulation with a non-overlapping shRNA elicited similar effects on sterol biosynthesis genes (Supplementary Fig. 8d).

We then performed gain-of-function studies and we stably expressed full-length *lincNORS* (NORS_OE) or vector-only control and performed RNA-seq on cells growing under ambient $O_2$ (Supplementary Fig. 9a and Supplementary Data 6). Both GSEA and IPA identified cholesterol biosynthesis among the top suppressed signatures (Supplementary Fig. 9b, c), with multiple components downregulated (Supplementary Fig. 9d). We confirmed by qPCR the downregulation of *MSMO1, SQLE*, and *HMGCR* as a result of *lincNORS* transcript induction (Supplementary Fig. 9e). As expected, this intervention did not significantly affect *miR-193b* or *miR-365a* levels (Supplementary Fig. 7b). Based on RNA-seq data, *lincNORS* downregulation and overexpression did not significantly affect the expression of flanking genes *MKL2/MRTFB* and *PARN*, thus indicating a predominantly *trans* activity (Supplementary Fig. 8e).

We found that the overall effect of *lincNORS* on total cellular cholesterol reflected the changes in transcript abundance. These were demonstrated in both loss- and gain-of-function studies, using two cell types (MCF7 and MDA-MB-231) and with two quantification methods (mass spectrometry and a commercial kit) (Fig. 4h and Supplementary Fig. 8g). Furthermore, the effect of *lincNORS* on cholesterol/lanosterol ratio supports the notion that it affects predominantly the distal products of the sterol/steroid biosynthesis than lower molecular weight precursors. The hydroxylated derivative 25-OH cholesterol did not appear to change significantly, consistent with the lack of response of the transcript for the conversion enzyme (Fig. 4i and Supplementary Fig. 8h).

**The impact of lincNORS on cellular phenotypes.** Functionally, the importance of sterol biosynthesis is not restricted to

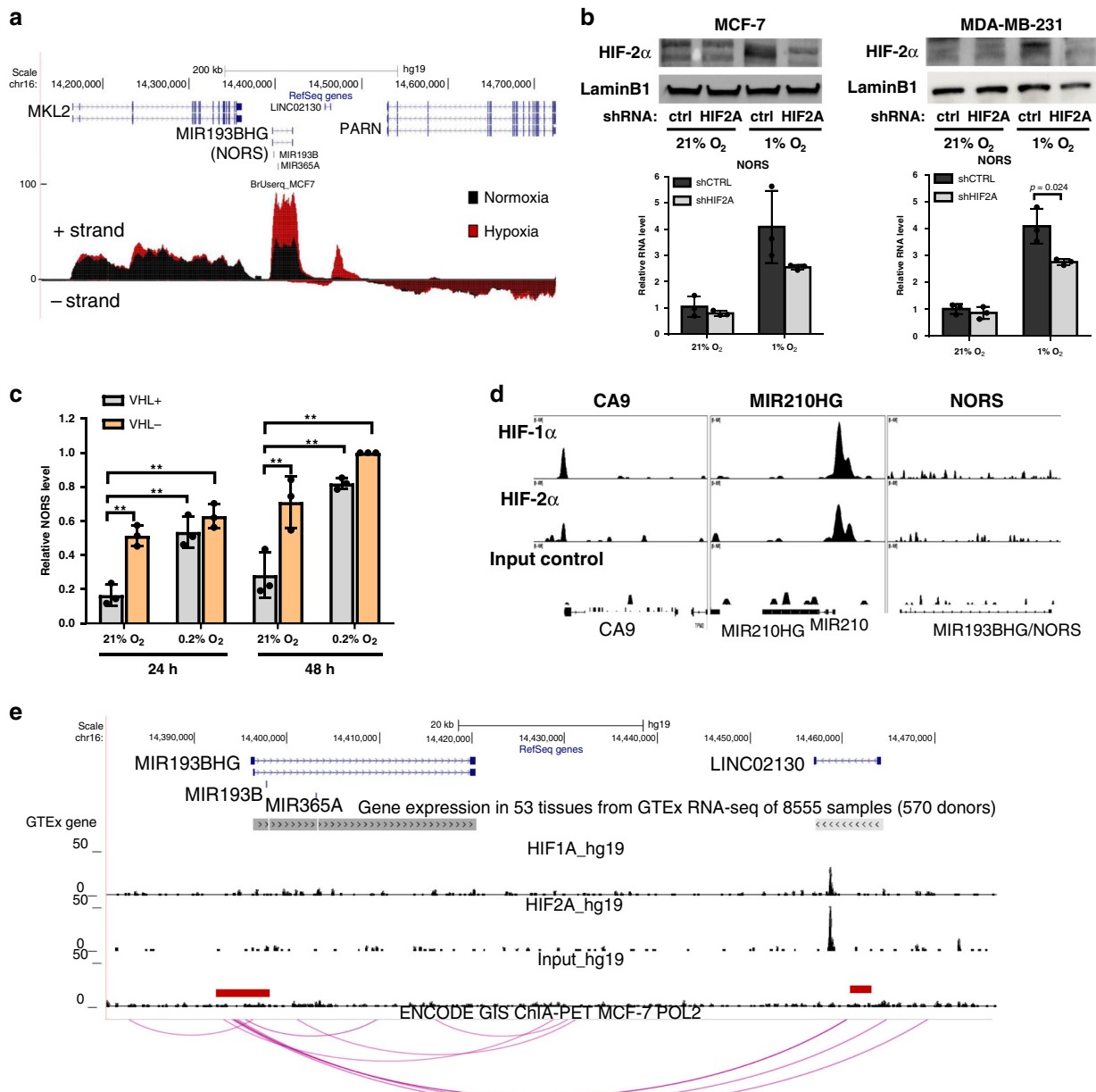

**Fig. 2 lincNORS expression is regulated by HIF-2α. a** UCSC genome browser view of newly synthesized RNA at the *lincNORS(MIR193BHG)* locus in MCF-7 cells cultured under 21% $O_2$ (black) and 1% $O_2$ (red) detected by Bru-seq. Strand-specificity was represented by positive/negative scales. **b** qPCR analysis of *lincNORS* expression in MCF-7 and MDA-MB-231 cells with stable HIF-2α knockdown. MCF-7 and MDA-MB-231 cells stably expressing HIF-2α shRNA or control shRNA were cultured in 21% $O_2$ or 1% $O_2$ for 48 h. HIF-2α knockdown was confirmed by western blot and representative results are shown. Data are shown as mean ± SD from three biological replicates (*$p < 0.05$, two-sided Student's *t*-test). **c** qPCR analysis of *lincNORS* expression in 786-O cells expressing empty vector (VHL−) or reconstituted wild type VHL (VHL+) cultured in 21% $O_2$ or 1% $O_2$. Data are shown as mean ± SD from three biological replicates (**$p < 0.01$, two-way ANOVA followed by Tukey's test). **d** UCSC genome browser view of HIF-1α and HIF-2α binding in *lincNORS*, *CA9*, and *MIR210HG* promoters in MCF-7 (ChIP-seq GSE28352). **e** UCSC genome browser view displaying HIF-1α and HIF-2α binding around *lincNORS* locus (MCF-7: ChIP-seq GSE28352), and chromatin interaction between *lincNORS* promoter and the HIF binding region within *LINC02130* (MCF-7: RNA PolII_ChIA-PET GSE33664).

hepatocyte physiology, being also essential for steroid hormone production, as well as for cellular growth and differentiation programs[34]. Furthermore, it is intricately connected to steroid hormone signaling and has been recently recognized as critical for neoplastic cell escape from steroid hormone dependence[35–38]. Given the diverse biological roles of this pathway, we hypothesized that the experimental manipulation of *lincNORS* would have consequences on phenotype.

To test this possibility, we examined the effects of *lincNORS* upregulation and downregulation in standard assays for cell viability and invasion under different environmental conditions. *lincNORS* knockdown had consistent pro-invasive consequences, with a probable contribution from deregulated sterol biogenesis, as suggested by the mitigating effect of simvastatin (Fig. 5a, b and Supplementary Fig. 10a, b). A similar, although more modest effect was observed in cell viability assays (Fig. 5c and Supplementary

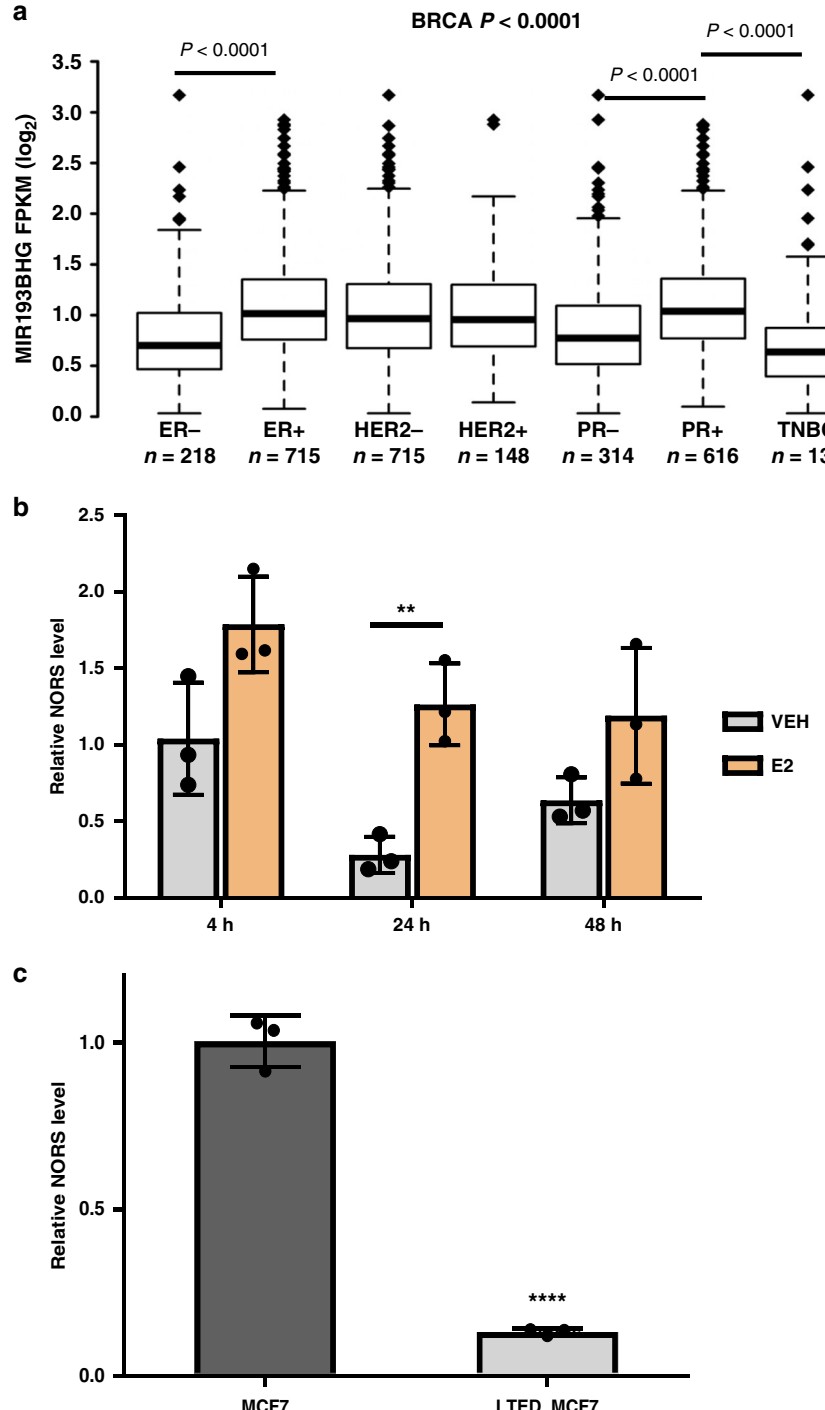

**Fig. 3 lincNORS is associated with estrogen response. a** *lincNORS/MIR193BHG* level in subtypes of breast cancer tumors in TCGA database. Boxplot represents first (lower bound), median (bar inside), and third (upper bound) quartiles, whiskers represent 1.5 times the interquartile range. Kruskal–Wallis test followed by Dunn's multiple comparison test. **b** qPCR analysis of *lincNORS* in MCF-7 cells treated with 10 nM E2 or vehicle control for the indicated time. Data are shown as mean ± SD from three biological replicates (**$p = 0.004$, two-sided Student's $t$-test). **c** qPCR analysis of *lincNORS* level in MCF-7 and long-term estrogen-deprived MCF7 cells. Data are shown as mean ± SD from three biological replicates (****$p < 0.0001$, two-sided Student's $t$-test).

Fig. 10c, d). Consistent evidence was observed in MDA-MB-231 xenografts with stably suppressed or overexpressed *lincNORS*. Knockdown resulted in increased metastatic load, whereas stable overexpression had the opposite effect (Fig. 5d, e). However, *lincNORS* status in xenografts did not significantly affect primary tumor size (Supplementary Fig. 10e). The effect of shRNA on *lincNORS* expression was still detectable in the harvested tumors,

while *miR-193b* expression remained unchanged (Fig. 5f). Steroid biosynthesis transcripts underwent changes similar to those observed in tissue culture (Fig. 5g).

**Mechanistic insights into *lincNORS* effects**. We next investigated whether *lincNORS* impacts steroid biosynthesis genes by

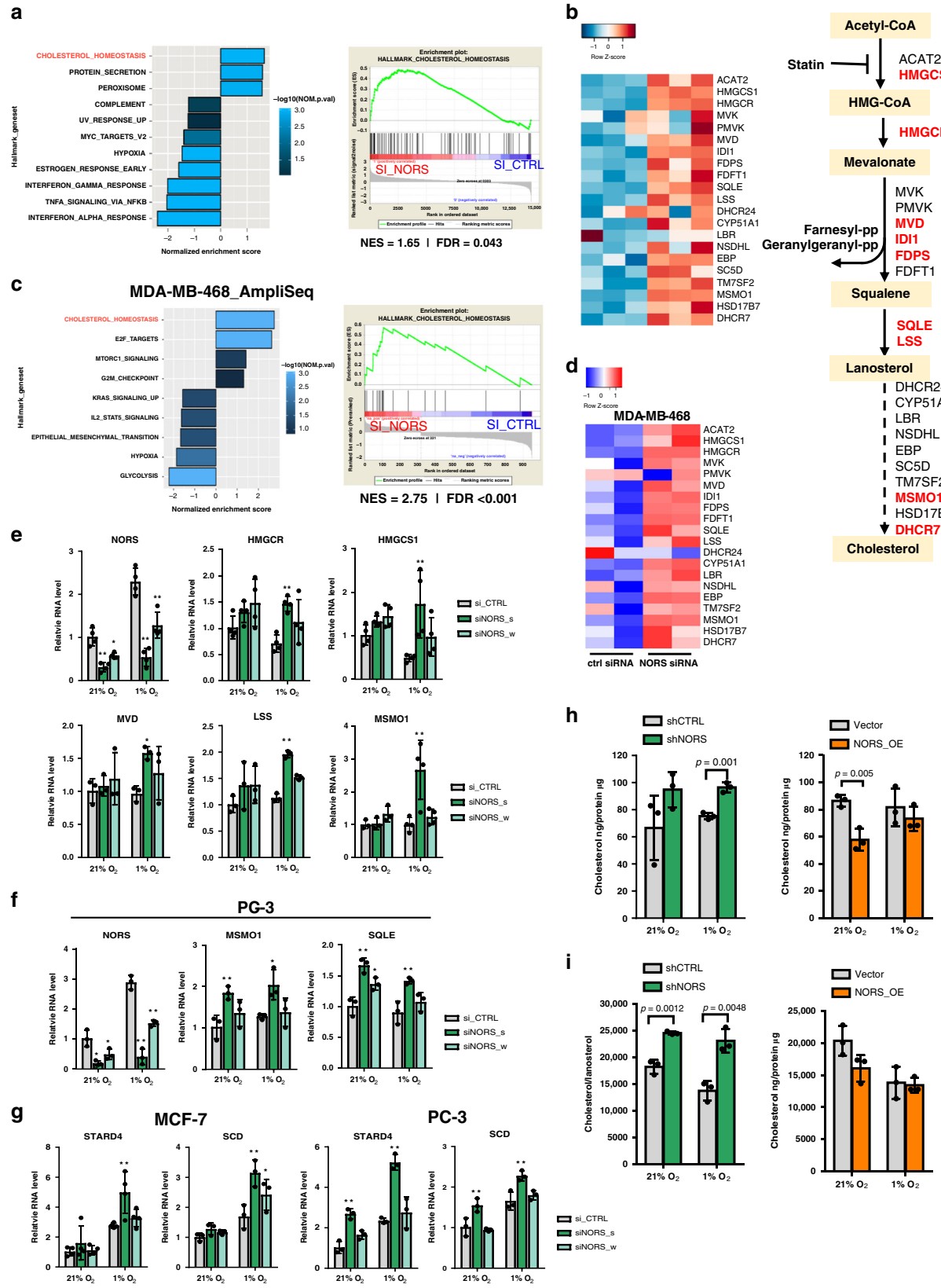

suppressing transcription or by altering mRNA turnover. Following siRNA-mediated *lincNORS* knockdown, nascent transcripts were labeled for 30 min with bromouridine (BrU), collected by immunoprecipitation with anti-BrU antibody, and quantified by qPCR analysis. Consistent with a specific posttranscriptional effect, siRNA treatment lowered total *lincNORS* transcript, while exhibiting no significant effect in the BrU-labeled pool (Supplementary Fig. 11a). We performed similar quantifications for steroid biosynthesis components and identified significant enrichment in BrU-labeled transcripts following *lincNORS*-depletion (Supplementary Fig. 11a).

**Fig. 4 lincNORS modulates cellular sterol homeostasis. a** Gene set enrichment analysis of hallmark gene sets associated with *lincNORS* knockdown versus the control in MCF-7 (FDR < 0.1). Enrichment plot of cholesterol homeostasis gene set is shown on the right. **b** Heatmap depicting expression of genes in the cholesterol synthesis pathway in MCF-7 cells transfected with *lincNORS* siRNA or siRNA control. The transcriptome was profiled by RNA-seq. A schematic diagram of the cholesterol synthesis pathway is shown on the right. Differentially expressed genes (FDR < 0.05) are marked with red. **c** Gene set enrichment analysis of hallmark gene sets associated with *lincNORS* knockdown versus the control in MDA-MB-468 (FDR < 0.1). Enrichment plot of cholesterol homeostasis gene set is shown on the right. **d** Heatmap depicting expression of genes in the cholesterol synthesis pathway in MDA-MB-468 cells transfected with *lincNORS* siRNA or siRNA control. The transcriptome was profiled by AmpliSeq. **e** qPCR analysis of selected cholesterol synthesis genes in MCF-7 cells transfected with two different *lincNORS* siRNA. Data represent mean ± SD from three/four biological replicates (*$p < 0.05$, **$p < 0.01$, two-way ANOVA with Dunnett's test). **f** qPCR analysis of selected cholesterol synthesis genes in PC-3 cells transfected with two different *lincNORS* siRNA. Data represent mean ± SD from three biological replicates (*$p < 0.05$, **$p < 0.01$, two-way ANOVA with Dunnett's test). **g** qPCR analysis of STARD4 and SCD expression in MCF-7 and PC-3 cells transfected with two different *lincNORS* siRNA. Data represent mean ± SD from three biological replicates (*$p < 0.05$, **$p < 0.01$, two-way ANOVA with Dunnett's test). **h** LC–MS measurement of total cholesterol in MDA-MB-231 cells with stable *lincNORS* knockdown or overexpression. Cholesterol reading was normalized to total protein content in each sample. Data represent mean ± SD from three biological replicates (*$p < 0.05$, **$p < 0.01$, ***$p < 0.001$, two-sided Student's *t*-test). **i** LC–MS metabolite measurement of cholesterol/lanosterol ratio in MDA-MB-231 cells with stable *lincNORS* knockdown or overexpression. Data represent mean ± SD from three biological replicates (*$p < 0.05$, **$p < 0.01$, ***$p < 0.001$, two-sided Student's *t*-test).

In contrast, we did not observe significant changes in transcript stability, suggesting a predominantly transcriptional impact of *lincNORS* (Fig. S11b).

To establish the mechanistic link between *lincNORS* and sterol biosynthesis, we searched for protein partners bridging this gap. To this end, we followed a well-established RNA affinity purification approach named MS2-TRAP, based on the tagging of *lincNORS* by addition of *MS2* RNA hairpins[39]. We collected complexes binding to MS2-*lincNORS* or control *MS2* RNA in MCF7 cells and subjected them to mass spectrometry, followed by a selection of proteins with no binding to the *MS2* RNA (Supplementary Fig. 12a and Supplementary Data 7). Among the high number of proteins identified, we did not find nuclear receptors directly or indirectly affecting sterol biogenesis (e.g., SREBP1, SREBP2, ESR1, NR5A1), but we observed many proteins implicated directly or indirectly in controlling gene expression programs (Supplementary Data 7). However, this data set, as a whole, was not validated and due to the experimental caveats of MS2 purification, may contain false positives. The candidates were further filtered based on several criteria: (1) documented RNA-binding protein status (RBP); (2) predominantly nuclear localization; (3) predicted to bind motifs within *lincNORS* by RBPmap[40]; and (4) documented connection to sterol/steroid pathways. Based on these criteria, we focused on two members of a known complex: RBP-associated with lethal yellow mutation (RALY) and its frequent partner heterogeneous nuclear ribonucleoprotein C (HNRNPC) (Fig. 6a). The former is of particular interest for multiple reasons: (1) a recent study in mice by Tontonoz' group described a role for *Raly* in the regulation of cholesterol biosynthesis, in partnership with a lncRNA termed *LeXis*[41]; (2) Nab3, a putative yeast homolog of *Raly* has been shown to regulate the ergosterol pathway[42,43]; (3) the RALY protein is located in the nucleus, predominantly in the chromatin-associated fraction, which is the same subcellular location of *lincNORS* (Fig. 6b); (4) lastly, *lincNORS* is present among the significantly enriched RALY-associated transcripts in MCF7 cells (under the designation RP11-65J21.3)[44]. As shown in Supplementary Fig. 12b, *lincNORS* is also among the top hypoxia-responsive lncRNAs bound by RALY.

We confirmed the interaction of endogenous *lincNORS* with RALY by RNA immunoprecipitation (RIP) analysis (Fig. 6c). The importance of RALY for the function of *lincNORS* was revealed by loss-of-function experiments. As shown in Fig. 6d, depletion of RALY using a specific siRNA compromised *lincNORS* regulatory effects on steroid biosynthesis genes. Reexamination of RALY RIP-seq data revealed that RALY interacts significantly with sterol biosynthesis transcripts (Fig. 6e). Experimentally, we

demonstrated that *lincNORS* can act as a modifier for some of these interactions. Specifically, *lincNORS* depletion measurably increased the association of RALY with *MSMO1* and *SQLE* (Fig. 6f). Again, this effect was more prominent at lower oxygen tension. These results suggest a dynamic interplay between RALY and the sterol biogenesis program in which *lincNORS* can act as a modifying factor. This strategy may reveal additional bona fide interactors from the candidate list and lead to a more complete image of *lincNORS* regulatory actions.

**Evidence of *lincNORS* biological relevance from GWAS.** Although phenotypic associations of naturally occurring genetic variations can provide uniquely valuable information about gene function, noncoding loci remain insufficiently interrogated using this approach. During the past decade, multiple independent genome-wide association studies (GWAS) have reported highly significant phenotypic associations for a cluster of single-nucleotide polymorphisms (SNPs) situated between the *MKL2* and *PARN* loci (Table 1 and Fig. 7a). Although these studies typically list *MKL2* as the name of the locus since it is the closest coding gene, we provide multiple arguments for *lincNORS* as the more plausible functional element.

From a biochemical perspective, most phenotypes listed in Table 1 are tied to the actions of sex steroids, thus plausibly affected by a modulator of sterol/steroid biosynthesis expressed mainly in hormone-responsive tissues[45–51]. Furthermore, an examination of the *MKL2-PARN* genomic region immediately revealed that these SNPs are more likely to impact *MIR193BHG/lincNORS* than *MKL2* (or *PARN*). First, the top GWAS-reported SNP for puberty onset (rs246185) is very close to the TSS of *lincNORS* and overlaps with histone acetylation/methylation peaks (Fig. 7a). Second, expression quantitative trait loci (eQTL) analysis in the GTEx database shows that these genetic variations are correlated with *lincNORS*, but not *MKL2* expression in several tissues (example shown in Fig. 7b). Colocalization analysis showed that the GWAS signal for age at menarche and the eQTL have a high probability of representing the same underlying signal in skeletal muscle and artery (Supplementary Data 8)[52,53]. Finally, based on ARIES mQTL information, these variants also exhibit significant associations with the methylation of the *MIR193BHG* locus rather than *MKL2*. Interestingly, these mQTLs are particularly robust during childhood and adolescence (Supplementary Fig. 13a), consistent with this locus being particularly important during a specific developmental stage (i.e., during childhood).

For a preliminary validation of these variants, we focused on a 1-kb fragment containing rs246185. This segment also contains

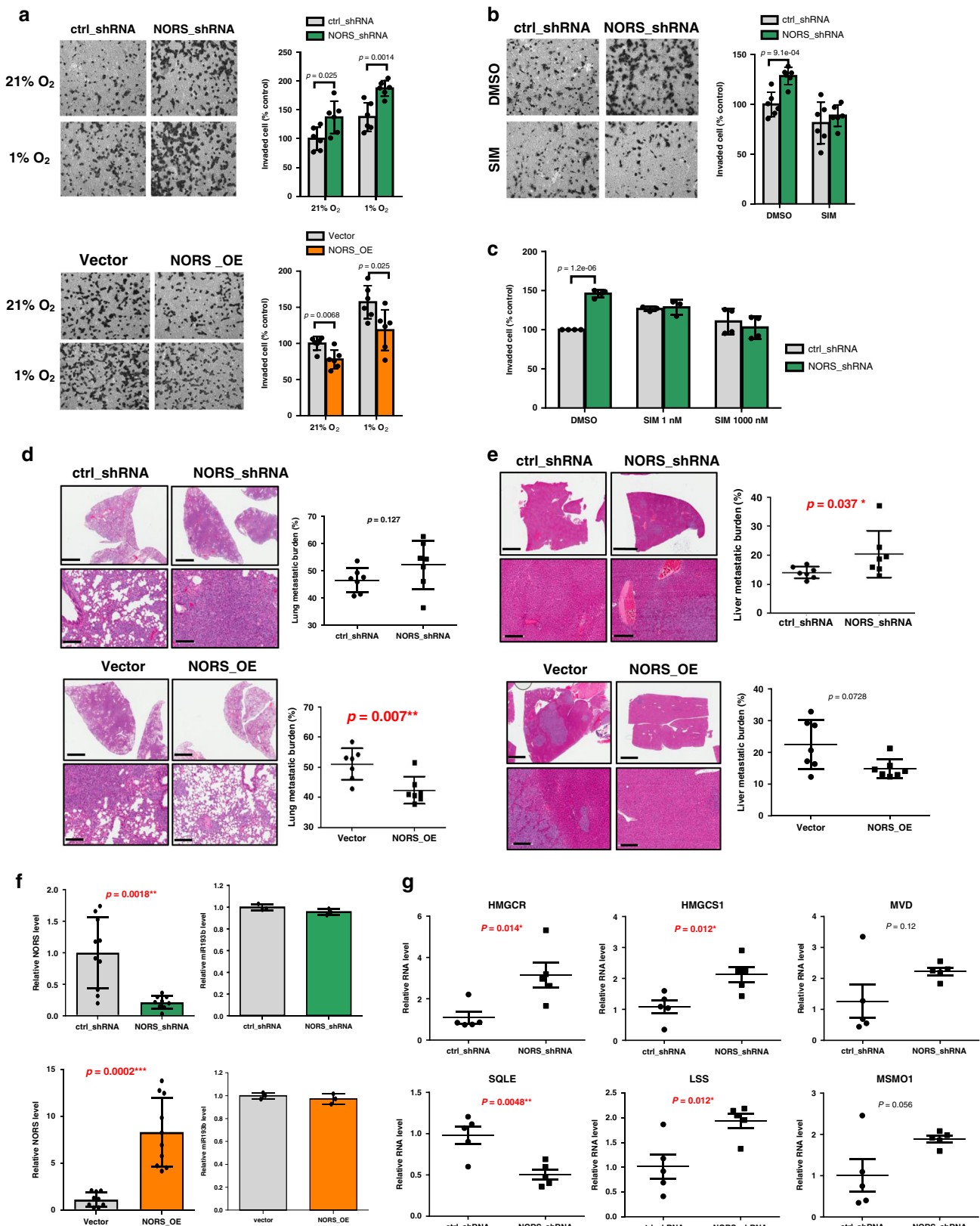

rs181766, which is in high linkage disequilibrium with rs246185 and exhibits similarly significant phenotypic associations according to the UK Biobank database (Fig. 8a). However, based on ChIP-seq and open chromatin by Formaldehyde Assisted Isolation of Regulatory Element (FAIRE-seq) data sets from ENCODE, rs181766 represents a potentially stronger causal

effector SNP candidate (Supplementary Fig. 13b). Sequences corresponding to the reference or alternative alleles of rs246185 (T/C) and rs181766 (T/C) were subcloned into pGL4 vectors and reporter activity was assayed in MCF7 cells. Although the Alt rs246185 (C) variant had no measurable impact, substitution of the rs181766-C allele for the T allele had a significant effect on

**Fig. 5 lincNORS inhibits cell invasion and metastasis of breast cancer cells. a** MDA-MB-231 cells with either knockdown or overexpression of *lincNORS* were plated in the Boyden chamber with Matrigel and cultured in 21% $O_2$ or 1% $O_2$. The cells that invaded through Matrigel were quantified 20 h after plating. Data are shown as mean ± SD from six biological replicates (two-sided Student's *t*-test). Representative images (×200 magnified) from three independent experiments were shown. **b** MDA-MB-231 cells with stable expression of *lincNORS* shRNA or control shRNA were plated in a Boyden chamber with Matrigel in the presence of 1 μM simvastatin or DMSO. The cells that invaded through Matrigel were quantified 20 h after plating. Data are shown as mean ± SD from three biological replicates (two-sided Student's *t*-test). Representative images (×200 magnified) from three independent experiments were shown. **c** MDA-MB-231 cells with stable expression of *MIR193BHG* shRNA or control shRNA were cultured in 1% $O_2$ for 48 h in the presence of simvastatin or DMSO. Cell viability was measured by MTT assay. Data are shown as mean ± SD from four independent experiments (two-sided Student's *t*-test). **d**, **e** Analysis of the metastasis of *lincNORS* knockdown or overexpressing MDA-MB-231 cells implanted in mice. Representative H&E-stained lung (**d**) and liver (**e**) sections are shown (bars in the upper panel, 3 mm; bars in the lower panel, 200 μm). The percentage of metastatic tumor burden over the total section area was analyzed using Aperio. Data are plotted as mean ± SD (two-sided Mann–Whitney *U* test, *N* = 7). **f** Expression of *HMGCR*, *HMGCS1*, *MVD*, *SQLE*, *LSS*, and *MSMO1* in tumor xenografts generated from MDA-MB-231 cells with stable *lincNORS* knockdown or control cells was measured by qPCR. Data were plotted as mean ± SD (two-sided Student's *t*-test). **g** *lincNORS* and *miR193b* expressions in the tumor xenografts were measured by qPCR. Data are plotted as mean ± SD (two-sided Student's *t*-test).

reporter activity (Fig. 8b). *lincNORS* exhibits the only significant eQTL for rs181766 in GTEx data sets, as described above for the neighboring GWAS SNPs.

## Discussion

Recent noncoding transcriptome studies have revealed new regulatory mechanisms for lipid metabolism based on lincRNAs such as *LncLSTR, HULC, H19, LeXis, MeXis*, and *MALAT1*[54]. Our study expands this group with another regulator of the steroid network program, termed *lincNORS*.

Understanding how physiological low $O_2$ impacts the homeostasis of sterols and steroid derivatives has broad implications on physiology and disease[55–58]. Accordingly, knowledge of tissue-specific modulators, such as lncRNAs, should be particularly informative. Although highly conserved between primates, the regions corresponding to the exons and surrounding regulatory elements of *lincNORS* exhibit rather limited conservation between the human and mouse genome. It could be argued therefore that the relevance of a mouse knockout model for human biology would be limited. On the other hand, naturally occurring genetic variations in human populations deliver a consistent message about this locus. As described above, single-nucleotide variations close to the TSS of *lincNORS* are significantly associated with sex hormone-related phenotypes such as the timing of puberty, height growth, and genital development. In this study, we mapped the GWAS signal to a plausible causal SNP and its target effector transcript.

How does the insight from GWAS come together in a cohesive model with *lincNORS* biochemistry? At this stage, we can only propose a simplistic scenario, with further details anticipated to emerge from future studies. For example, the variant of the minor allele could trigger increased baseline *lincNORS* expression, leading to more potent suppression of steroid biosynthesis in hormonally-responsive tissues and ultimately to a delay in sexual maturation. Based on the directionality of eQTLs and the nature of transcriptional regulators (both negative and positive) recruited in the proximity of rs181766, the causal alternative allele may exert a positive or negative impact on *lincNORS* expression, depending on tissue context.

With the phenotypic connection in mind, it only makes sense that *lincNORS* responds to estrogen treatment and that its genomic neighborhood appears to be a hub for hormone receptor recruitment. How does hypoxia, the original context that brought *lincNORS* to our attention, fit into this context? In recent years the importance of HIFs and physiologically low $O_2$ for reproductive organ physiology is coming to light from multiple directions[59–62]. In summary, one could imagine *lincNORS* as an integrator of microenvironmental and hormonal cues that modulates key aspects of sterol/steroid homeostasis that are essential for growth and development.

In hindsight, *lincNORS*' participation in both low oxygen and estrogen response should not come as a surprise, as these networks are increasingly recognized as interconnected[13]. Evidence for a similar dual facet already exists for other hypoxia-inducible lncRNAs such as *NEAT1* and *MALAT1*[63,64]. In another similarity to *lincNORS*, in hypoxic cells, it was previously demonstrated that *NEAT1* and MALAT1 were controlled predominantly by HIF-2, recruited to distal enhancer elements rather than proximal promoter[6].

Our study may also expand the emerging link between RALY and sterol/steroid biosynthesis[41]. It is plausible that the specific noncoding elements alter the function of RALY-containing complexes depending on the cellular context. For example, in hepatocytes in interaction with *LeXis* RALY becomes relevant for the "classic" cholesterol biosynthesis. On the other hand, by partnering with *lincNORS* in hormonally-responsive tissues RALY becomes involved in the extrahepatic facets of this pathway[41]. As RALY has the potential to regulate gene expression at multiple steps (positively or negatively), noncoding RNA interactors may be critical for action specificity[44,65].

## Methods

**Cell culture and reagents**. Human cancer cell lines MCF-7, MDA-MB-231, MDA-MB-468, T47D, A549, HT-29, SKOV3, MIA-PaCa2, and U118 were purchased from American Type Culture Collection (ATCC) and maintained at 37 °C in 5% $CO_2$ with DMEM/High glucose medium supplemented with 10% (v/v) FBS (Biowest) and antibiotics. All cell lines were routinely tested for mycoplasma contamination (MycoAlert™ Mycoplasma Detection Kit, Lonza). Long-term estrogen-deprived (LTED) MCF7 cells were developed as previously described[66]. Briefly, MCF7 cells were maintained in medium with 10% dextran-coated charcoal-stripped FBS (Hyclone Laboratories, Inc., Logan, UT) for 6 months to generate LTED cells. Renal clear cell carcinoma cells 786-O with empty vector (VHL−) and 786-O with reconstituted VHL (VHL+) were obtained from Dr. Maria Czyzyk-Krzeska (University of Cincinnati). Patient-derived pancreatic cancer cell lines Pa03C, Panc10.05, Panc198, as well as cancer-associated fibroblast cell line CAF19, were obtained from Dr. Anirban Maitra at Johns Hopkins University[67]. Human lung fibroblast cell line WI-38 was obtained from Dr. Shadia Jalal's lab (Indiana University School of Medicine). Fallopian tube FTE282 epithelial cells (a gift from Dr. Ronny Drapkin, University of Pennsylvania) were cultured in DMEM-Ham's F12 media (Corning) supplemented with 2% Ultroser G (Crescent Chemical Company) by Dr. Sumegha Mitra. Hypoxia cell culture was performed at 0.2%, 1%, or 2% $O_2$ for the indicated time in an InVivo200 hypoxia workstation (Ruskin, Inc., Cincinnati, OH). Dimethyloxalylglycine (DMOG, Frontier Scientific D1070-100 mg) was dissolved in DMSO and used at 1 mM. Simvastatin (Sigma S6196) was used at indicated concentrations. Estradiol (Sigma E8875) was dissolved in ethanol and used at 10 nM. In estradiol experiments, cells were plated and maintained in medium with 10% dextran-coated charcoal-stripped FBS. Actinomycin D (Sigma A1410) was used at 2.5 μg/ml.

**siRNA transfection**. Silencer®Select siRNAs were purchased from ThermoFisher Scientific and transfected at 10 nM final concentration using RNAiMAX reagent (Invitrogen, CA) with a reverse transfection protocol. siRNAs used in the study

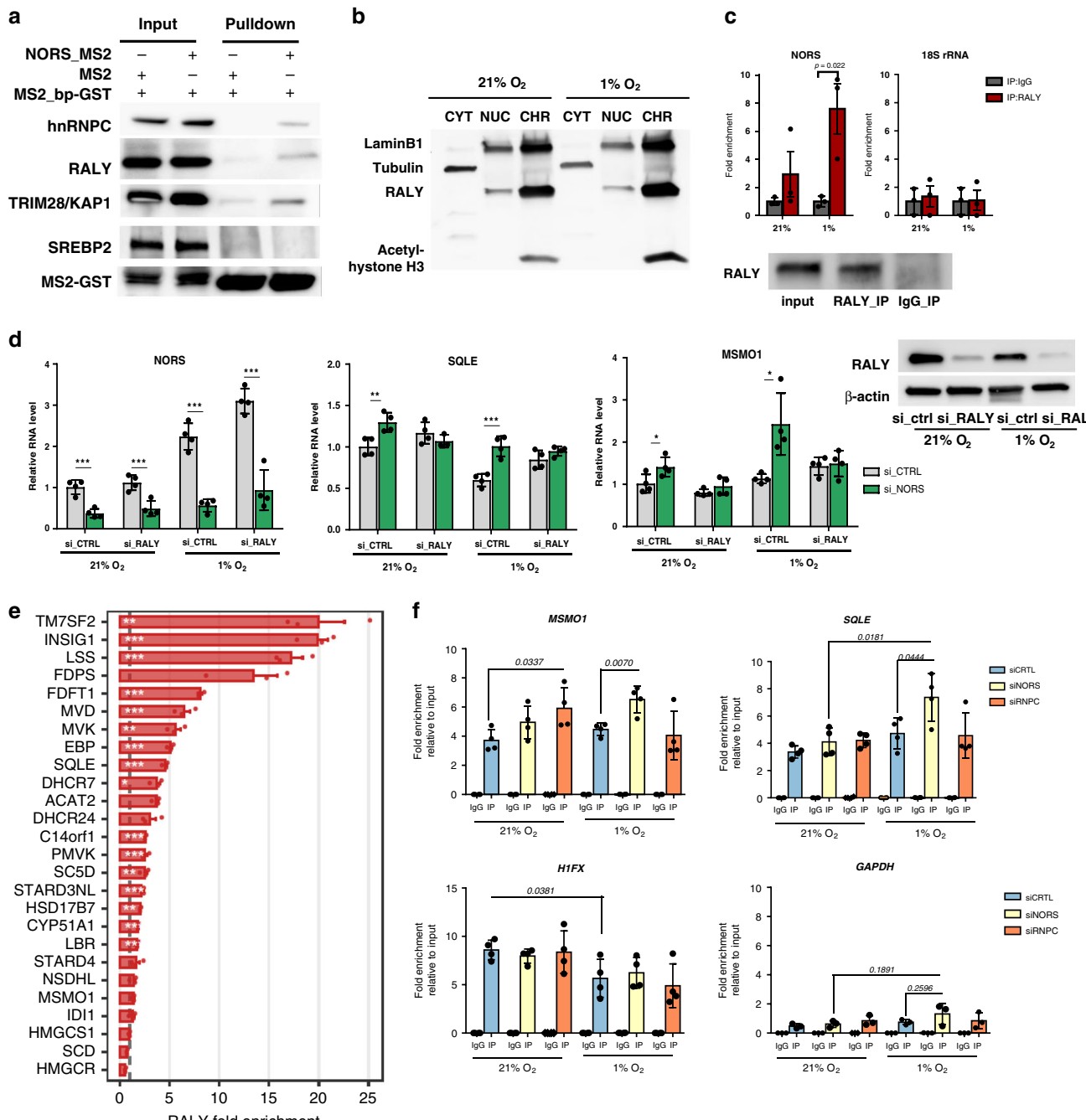

**Fig. 6 lincNORS regulates cholesterol synthesis via interaction with RALY. a** The level of hnRNPC, RALY, TRIM28, SREBP2, and MS2-GST in the MS2 pulldown material and input lysate was detected by western blot. The data presented are a representative image from two to three independent experiments. **b** Subcellular location of RALY was detected by western blot. Tubulin, Lamin-B1, and acetyl-histone H3 were included as controls for cytoplasmic, nuclear, and chromatin-associated fractions. A representative image from three independent experiments is shown. **c** RNA immunoprecipitation (RIP) analysis of RALY. The presence of *lincNORS* and 18s rRNA in the precipitated complex was detected by qPCR. Data represent mean ± SD from three biological replicates (two-sided Student's *t*-test). **d** qPCR analysis of *NORS, MSMO1*, and *SQLE* expression in MCF-7 cells transfected with control, *lincNORS* siRNA, and/or *RALY* siRNA. Data represent mean ± SD from four biological replicates (two-sided Student's *t*-test). RALY knockdown was confirmed by western blot in each experiment and a representative image is shown on the right. **e** Enrichment of RNA of cholesterol synthesis genes in RALY-containing immunoprecipitated complex. The barplot displays the mean fold enrichment ± S.E.M. from three independent RALY RIP-Seq vs control experiments in MCF7 cells. The statistical significance of the enrichment was determined with CuffDiff v2.2.1 (*FDR < 0.05, **FDR < 0.01, ***FDR < 0.001). Source data are provided as a Source Data file. **f** Enrichment of *MSMO1, SQLE, H1FX,* and *GAPDH* in RALY-containing immunoprecipitated complex after *lincNORS* or *hnRNPC* knockdown was detected by qPCR. Data represent mean ± SD from four biological replicates (two-sided Student's *t*-test).

**Table 1 Summary of genome-wide association studies reported SNPs in *lincNORS* promoter.**

| SNP.Id | CHR | POS_hg38 | Distance[a] | GWAS MAPPED_TRAIT | RISK.ALLELE | RAF[b] | *p*-value | Beta | Pubmed id | Journal |
|---|---|---|---|---|---|---|---|---|---|---|
| rs1659127 | 16 | 14294448 | 7840 | Age at menarche | rs1659127-A | 0.34 | 4.00E−09 | 2.4[c] | 21102462 | *Nat Genet* |
| rs1704528 | 16 | 14294893 | 7395 | Age at menarche | rs1704528-? | NR | 6.00E−12 | −0.033 | 27182965 | *Nat Genet* |
| rs246185 | 16 | 14301575 | 713 | Age at menarche | rs246185-C | 0.33 | 6.80E−16 | 0.04 | 25231870 | *Nature* |
| | | | | Tanner puberty stage | rs246185-T | 0.68 | 8.88E−09 | 0.141 | 24770850 | *Hum Mol Genet* |
| | | | | Age at voice drop | rs246185-? | NR | 7.10E−14 | −0.044 | 27182965 | *Nat Genet* |
| rs1704529 | 16 | 14294769 | 7519 | Androgenetic alopecia | rs1704529-C | NR | 1.42E−09 | −0.090334 | 29146897 | *Nat Commun* |
| rs246180 | 16 | 14298066 | 4222 | Androgenetic alopecia | rs246180-? | NR | 8.00E−09 | 0.141 | 27182965 | *Nat Genet* |
| rs246185 | 16 | 14301575 | 713 | Androgenetic alopecia | rs246185-C | 0.33 | 2.65E−10 | 0.14 | 28272467 | *Nat Commun* |
| rs1659127 | 16 | 14294448 | 7840 | Body height | rs1659127-A | 0.48 | 6.70E−10 | 0.041 | 25429064 | *Hum Mol Genet* |
| | | | | Body height | rs1659127-A | 0.34 | 1.10E−11 | 0.027 | 20881960 | *Nature* |
| rs246185 | 16 | 14301575 | 713 | QT interval | rs246185-C | 0.34 | 3.00E−13 | 0.72 | 24952745 | *Nat Genet* |

[a]Distance: distance from SNP to *lincNORS* transcription start site.
[b]RAF: risk allele frequency.
[c]Per allele change in age at menarche (weeks).

include: Silencer®Select negative control siRNA (4390843), *FOSL2* siRNAs (Supplementary Data 9), *RALY* siRNA (s22655), NORS_siRNA_s (5′-GAGCGU-GUAUAAAACCAAAtt-3′), NORS_siRNA_w (5′-GCAAAGAUGUUUCCAGAGAtt-3′) and hnRNPC siRNA (5′-CAACGGGACUAUUAUGAUAdTdT-3′).

**Stable cell line generation and RACE**. The shRNA for *lincNORS* was designed to target the sequence 5′-GATTAAAGCAACATGTTATTC-3′. The hairpin template oligonucleotides were synthesized by Integrated DNA Technologies and subsequently cloned into lentiviral vector pLKO.1-puro. The scramble shRNA control was purchased from Sigma-Aldrich (SHC002). Lentivirus containing shRNA were generated in 293T cells. MDA-MB-231 and MDA-MB-468 cells were infected with the lentivirus and selected with 1 μg/mL puromycin (Gibco). All shRNA stable expressing cell lines were maintained with medium containing 1 μg/mL puromycin in subsequent cultures. 5′ and 3′ RACE were performed with MCF-7 cells using FirstChoice RLM-RACE Kit (Ambion) according to the manufacturer's instruction. For overexpression, *lincNORS* was amplified from cDNA using primers: forward: 5′-CGG GAT CCG TCG GCT GCG CGC TCT CTG-3′; reverse: 5′-GCT CTA GAC TCA ATA AAA TGC TGC TTA TTT ATT-3′ and subsequently cloned into pcDNA3.1 vector. Cells were transfected with empty vector or *lincNORS* expressing construct using Lipofectamine®2000 according to the manufacturer's instruction and selected with 1.0 mg/mL G418 (CORNING, 61234RG). The overexpression stable cell lines were maintained with a medium containing 0.5 mg/mL G418 in subsequent cultures.

**Bioinformatic analyses of public data sets**. Data analysis of level 1 TCGA RNA-seq data was performed under NIH dbGAP approval, Project #5187: "Landscape of tumor hypoxia in the TCGA data set (P.I. M. Radovich)". The expression of coding and noncoding genes was calculated using NGSUtils[68]. Expression correlation (Spearman) between *lincNORS* and each protein-coding gene was calculated using R (version 3.0.1). Hypoxia score of tumors was calculated using hypoxia metagene proposed by FM Buffa et al.[20] following the method formulated by Maud H.W. Starmans et al.[69]. HIF ChIP-Seq data (GSE28352) were used to assess whether HIF was directly involved in *lincNORS* induction by hypoxia. Raw data of HIF-1α ChIP-Seq, HIF-2α ChIP-Seq, and pre-immune ChIP-seq control were downloaded and aligned to hg19 using Bowtie (version 1.2.1.1). The peaks were visualized in UCSC genome browser. The other ChIP-Seq data presented in this study were obtained from ChIP-Altas (http://chip-atlas.org/) and visualized in IGV. Long-range chromatin interaction in MCF-7 was determined by ChIA-PET data published by Li et al. (GSE33664) and visualized in WashU EpiGenome Browser[26].

**Subcellular fractionation**. Subcellular fractionation was performed as described[70]. Briefly, cells were pelleted and resuspended in ice-cold HLB buffer (10 mM Tris pH 7.5, 10 mM NaCl, 3 mM MgCl$_2$, 0.3% (v/v) NP-40, 10% (v/v) glycerol) supplemented with either RiboLock RNase Inhibitor (Thermo #EO0381) or Halt Protease Inhibitor (Thermo #78442) for RNA and protein samples respectively. After 10 min incubation on ice, samples were centrifuged at $1000 \times g$ at 4 °C for 3 min and the supernatant was collected as the cytoplasmic fraction. The remaining pellet was washed with supplemented HLB buffer three times. Pellet was resuspended in ice-cold MWS buffer (10 mM Tris–HCl pH 7.0, 4 mM EDTA, 0.3 M NaCl, 1 M urea, and 1% (v/v) NP-40) supplemented with either RNase inhibitor or protease inhibitor (as above). After 15 min on ice, samples were centrifuged at 1000 × g at 4 °C for 3 min and the supernatant was collected as the nucleoplasmic fraction. The remaining pellet, the chromatin-associated fraction, was washed with

supplemented MWS buffer three times. For the RNA samples, 700 μL of QIAzol was added to resuspend the pellet. For protein samples, 250 μL of NLB buffer (20 mM Tris (pH 7.5), 150 mM KCl, 3 mM MgCl$_2$, 0.3% (vol/vol) NP-40, and 10% (vol/vol) glycerol) were added to the pellet, then sonicated three times at 20% power for 15 s in an ice bath with 2 min of cooling between cycles. After centrifugation, supernatants were collected.

**RNAscope**. Detection of *lincNORS* expression and subcellular location in MCF-7 cells with *lincNORS*-specific RNAscope® probe (designed by Advanced Cell Diagnostics, ACD) were performed using RNAscope® Multiplex Fluorescent Reagent kit (ACD 320850) according to the manufacturer's instructions. A negative control probe (ACD 322340) was also purchased from ACD. Images were visualized with Olympus BX43 microscope equipped with Olympus U-RFL-T as a power supply unit.

**Invasion assay**. Invasion assays were performed in a 24-well plate with BD Falcon cell culture inserts (Corning 353097) coated with Matrigel (BD Bioscience 356237). Cells were starved in serum-free medium overnight prior to the experiment. At the start of the assay, cells were washed with serum-free medium and plated into the upper chamber at $1 \times 10^5$ cells per well in 0.5 mL serum-free DMEM medium. Complete medium (0.75 mL) with 10% FBS was added to the lower chamber. Cells were allowed to invade for 20 h under either normoxic or hypoxic conditions. After incubation, the medium in the upper chamber was aspirated and non-invaded cells on the upper surface of the filter were removed with a cotton swab. The transwell membranes were fixed and stained with 0.25% crystal violet in formalin: methanol (1:9) solution for 30 min. After staining, the membranes were washed three times with water and air-dried. Invaded cells were visualized using light microscopy. 150 μL 2% SDS was then added to the membrane, incubated for 20 min to solubilize the crystal violet from stained invaded cells, then transferred to a new 96-well plate. The relative invasion was determined using absorbance readings at 550 nm.

**Cell viability assay**. Cell viability was measured by MTT assay. Cells were plated in 96-well plates (10,000–20,000 cells per well) and relevant treatment (siRNA transfection or drug treatment) was performed as described in each experiment. 4 h prior to the endpoints, 10 μL of MTT (5 mg/mL in PBS, Sigma-Aldrich M5655) were added to each well containing 100 μL medium and incubated for 4 h. The medium was aspirated and 100 μL acidic isopropanol (0.04 N HCl in isopropanol) was added to dissolve the insoluble purple formazan product into a colored solution. Absorbance was recorded at 570 nm, with a reference at 630 nm serving as the blank.

**RNA extraction and qPCR analysis**. For RNA extraction, cells were lysed using QIAzol Lysis Reagent (QIAGEN) and total RNA was isolated using the RNeasy kit (QIAGEN) following the manufacturer's instructions. Reverse transcription was performed using TaqMan® Reverse Transcription Reagent (Applied Biosystems, CA) from 1 μg of RNA, and qPCR was performed using gene-specific primers (Supplementary Data 9) and SYBR green master mix (Applied Biosystem, CA). *18 S* rRNA or *B2M* RNA level was used for normalization. For microRNA analysis, reverse transcription was performed with TaqMan® MicroRNA Reverse Transcription Kit using 10 ng RNA, and qPCR was performed using TaqMan® Gene

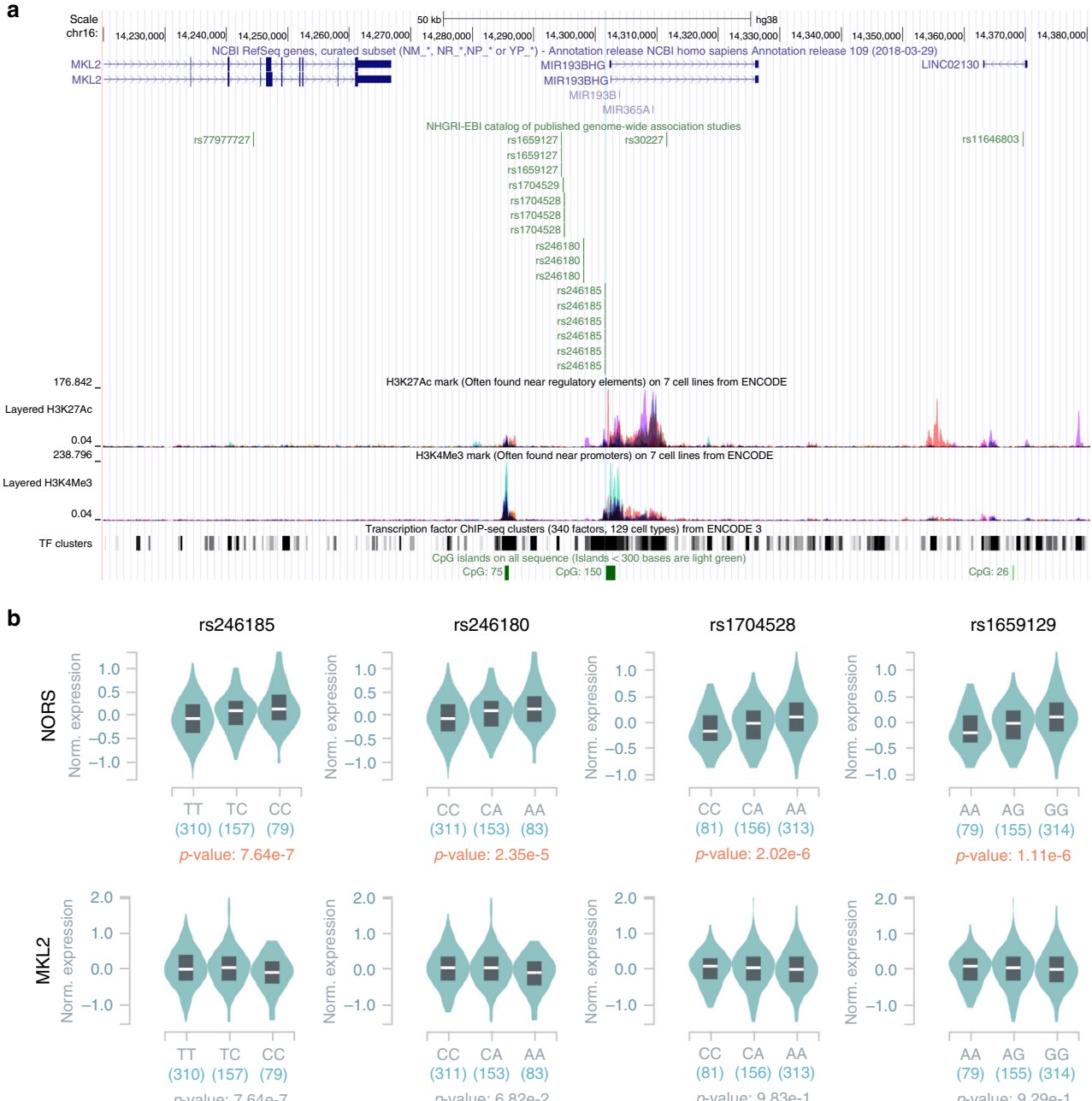

**Fig. 7 GWAS-reported SNPs in *lincNORS* promoter support its roles in steroid regulation. a** Locations of SNPs in *lincNORS* promoter reported in NHGRI-EBI catalog of published genome-wide association studies were visualized in UCSC genome browser. **b** eQTL boxplot showing genotype-expression relationships between rs246185, rs246180, rs1704528, and rs1659129 with *lincNORS* (top) or the closest protein-coding gene *MKL2* (bottom) in skeletal muscle.

Expression Master Mix. *U6* or *RNU48* expression was used as an endogenous control.

**Xenograft model.** All animal experiments were approved and carried out in accordance with the Institutional Animal Care and Utilization Committee at Indiana University School of Medicine. $1 \times 10^6$ cells were injected into the mammary fat pad of 10 ~4 weeks old female NSG (NOD.Cg-Prkdc^scid Il2rg^tm1Wjl/SzJ) mice. Tumor growth was monitored twice a week by measurement of the long and short diameters of the tumor using digital calipers. At the end of the experiments, animals were killed and the primary tumor, lung, and liver were resected. Lung and liver tissue were formalin-fixed, paraffin-embedded, sectioned, and stained with hematoxylin and eosin (H&E) for analysis of metastatic burden. The percentage of lung or liver tissue containing tumor metastasis was analyzed using Aperio (Leica Biosystems).

**RNA-seq and data analysis.** The initial RNA-seq used for identifying hypoxia-induced lncRNAs was performed on Illumina NextSeq500 platform. Sequenced reads were mapped to genome with TopHat2 version 2.1.1. Read counts for each gene were generated with HTSeq-Count from the HTSeq package version 0.6.1p1 and differential expression analysis was carried out using the DESeq2 package (version 1.12.3) in R/Bioconductor (R version 3.3.1)[71]. For analysis of *lincNORS* function, RNA sequencing was performed using The Ion PI Sequencing 200 Kit v2 Kit (4485149, Life Technologies) on an Ion Proton semiconductor sequencer according to the kit instructions using the Ion PI Chip Kit v2 (4482321, Life Technologies). Reads were aligned using STAR (version 2.3.0) and differential expression was analyzed using the EdgeR (version 3.8.6) package in R (version 3.0.1)[72,73]. Differential expression with FDR < 0.05 was considered significant.

**AmpliSeq and data analysis.** Automated library preparation was performed according to the protocol available with the Ion Chef (MAN0013432) and using the

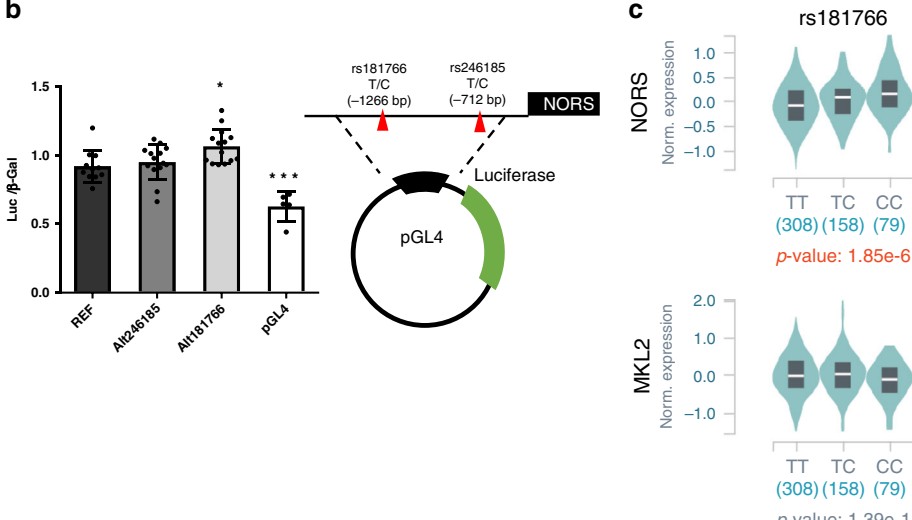

**a**

| Study ID | Trait | P-value | Beta |
|---|---|---|---|
| NEALEUKB_2375 | Relative age of first facial hair | 3.25E-36 | 0.0214554 |
| NEALEUKB_50 | Standing height | 1.17E-21 | 0.0178197 |
| NEALEUKB_2395_4 | Hair/balding pattern: pattern 4 | 2.81E-21 | −0.094798004 |
| NEALEUKB_2714 | Age when periods started (menarche) | 1.69E-19 | 0.0241905 |
| NEALEUKB_2385 | Relative age voice broke | 9.59E-18 | 0.0109283 |
| NEALEUKB_23129 | Trunk fat-free mass | 1.11E-12 | 0.0117562 |
| NEALEUKB_23130 | Trunk-predicted mass | 1.52E-12 | 0.0116464 |
| NEALEUKB_23121 | Arm fat-free mass (right) | 1.13E-10 | 0.0105375 |
| NEALEUKB_23101 | Whole body fat-free mass | 5.16E-10 | 0.0103055 |
| NEALEUKB_23102 | Whole body water mass | 7.65E-10 | 0.0102176 |

**Fig. 8 The impact of rs181766 on lincNORS promoter activity. a** List of physical traits that are reported as significantly associated with rs181766 in UK Biobank (https://genetics.opentargets.org/). **b** The effect of two selected SNPs rs246185, rs181766 on *lincNORS* expression was examined using luciferase assay. A diagram displaying the *lincNORS* promoter cloning was shown on the right. Data represent mean + SD from 4 to 11 biological replicates (*$p < 0.05$, ***$p < 0.005$, one-way ANOVA followed by Dunnett's multiple comparisons test against REF). **c** eQTL boxplot showing genotype-expression relationships between rs181766 and *lincNORS* (top) or the closest protein-coding gene *MKL2* (bottom).

Ion AmpliSeq Kit for Chef DL8 (A29024). Library was further diluted to 50 pM for sequencing. Template preparation and sequencing were executed using the Ion PI Hi-Q Chef Kit (A27198, ThermoFisher Scientific) with template quantification using the IonSphere Quality Control Kit (4468656, ThermoFisher Scientific). The raw read counts are included in the Supplementary Data 5. Standard workflow for gene-level expression was conducted using Torrent Suite Version 4.4. The AmpliSeq Transcriptome Plugin was used to generate standard expression files. Differential expression was performed in R using the DESeq2 package (version 1.12.3) in R/Bioconductor (R version 3.3.1). Differential expression with FDR < 0.05 was considered significant. Pathway analysis was performed using IPA and GSEA.

**Gene set enrichment analysis**. Gene set enrichment analysis was performed following GSEA User Guide (The Broad Institute)[74]. Briefly, each condition was considered as a group and gene list was ranked with GSEA default ranking metrics. Gene sets from the Molecular Signature Database were used in the analysis to identify the pathways significantly enriched in each group. Gene sets were permutated 1000 times to obtain empirical FDR corrected p-values.

**Western blot analysis**. Cell lysates were prepared in RIPA buffer containing protease inhibitor cocktail (ThermoFisher Scientific). For HIF western blot, cells were fractionated using RLN buffer (10 mM Tris–HCl pH 8.0, 140 mM NaCl, 1.5 mM MgCl₂, 0.5% NP-40, proteinase inhibitor cocktail) and the nuclear fraction was collected for analysis. Proteins were separated by SDS-polyacrylamide electrophoresis (SDS-PAGE), then transferred onto PVDF membranes (Biorad). Primary antibodies used in this study included those that recognized HIF-1α (R&D AF1935), HIF-2α (R&D AF2886; Cell Signaling #7096), β-actin (Thermo MA5-15739; Santa Cruz sc-1615), β-tubulin (Santa Cruz sc-5274), acetyl-histone H3 (Millipore 06-599), Lamin-B1 (Abcam ab16048), SREBP2 (R&D AF7119), hnRNPC (Santa Cruz sc-32308), TRIM28 (Abcam ab10483), RALY (Abcam ab170105), and Enterobacterio Phage MS2 coat protein (Millipore ABE76). For a complete list of antibodies used please see Supplementary Data 10.

**Total cholesterol quantification**. Cellular cholesterol was measured using the Wako Cholesterol assay kit (WAKO, #439-17501) and liquid chromatography-mass spectrometry. For the measurement using Wako Cholesterol assay kit, cellular lipids were extracted using 750 μL chloroform: methanol (2:1) solution. After extraction for 30 min on a rocker, samples were centrifuged for 10 min at $1500 \times g$ at 4 °C, then the lower chloroform phase was collected and air-dried in a fume hood until all liquid evaporated. The lipids were resuspended in 10% Triton-X100 in isopropanol. Total cholesterol content was measured using the Wako Cholesterol assay kit (WAKO, #439-17501) according to the manufacturer's instructions. For the measurement using liquid chromatography-mass spectrometry, Oxysterols were extracted from cells using a modified sterol extraction method from McDonald et al.[75]. Briefly, cell pellets were extracted with 1 mL of dichloromethane/methanol 50/50 (v/v) containing 5 μg of butylated hydroxytoluene (BHT) and 200 ng of each deuterated internal standard. 0.2 mL of water were added after mixing and vortexed for 15 s and centrifuged. The organic layer was transferred to a new tube and saponified with 0.6 mL of ethanolic KOH (1 N). The tubes were flushed with nitrogen, capped, and heated to 35 °C for 1 h. 2 mL of water were added and 2 mL dichloromethane. The aqueous fraction was extracted twice with dichloromethane then the two organic fractions were combined and dried with nitrogen gas at room temperature. The samples were then derivatized according to Honda et al.[76]. The derivatization reaction mixture was composed of 100 mg of 2-methyl-6-nitrobenzoic anhydride, 30 mg of 4-diethylaminopyridine, 80 mg of picolinic acid, 1.5 mL of tetrahydrofuran (THF). 0.15 mL of this mixture were added to each sample along with 0.02 mL of triethylamine (TEA). The reaction proceeded at room temperature for 1 h, then the samples were dried and subsequently reconstituted in 0.1 mL of acetonitrile just prior to LC/MS/MS analysis. The analysis was done with an Agilent 1200 liquid chromatography system coupled to an Agilent 6460 QQQ mass spectrometer (Santa Clara, CA) in Bindley Bioscience Center at Purdue University. Data were processed using Agilent Masshunter Quantitative analysis software (V.B.06).

**Bromouridine (BrU)-labeled RNA Immunoprecipitation and Bru-Seq**. BrU labeling of newly synthesized RNA and BrU-RNA isolation were performed following the method established by Ljungman's group[22]. Briefly, MCF-7 cells

transfected with either control siRNA or *lincNORS* siRNA were cultured in hypoxic conditions. BrU (Sigma-Aldrich 850187) at a final concentration of 2 mM was added to the plate when cells reached ~80% confluency. After labeling for 30 min, cells were harvested for total RNA extraction. BrU-labeled RNA was immuno-precipitated from the total RNA using anti-BrdU antibody (BD Pharmingen, 555627) conjugated with Dynabeads® Goat anti-mouse IgG beads (Invitrogen #11033). After 1-h incubation, beads were washed three times with 0.1% BSA in DEPC-PBS to remove unbound RNA then heated at 95 °C for 10 min to recover the BrU-RNA from the beads. The BrU-RNA and the input RNA were analyzed using qPCR. The same BrU-RNA isolation procedure was performed with MCF-7 cells cultured in 21% or 1% $O_2$ and the BrU-labeled RNA was sequenced at the University of Michigan Sequencing Core using Illumina HiSeq2000 sequencer.

**MS2-tagged RNA affinity purification**. RNA pulldown using MS2-tagged RNA affinity purification was performed according to the previously described protocol with slight adaptation[77]. *lincNORS* cDNA was cloned into the pMS2 vector upstream of the MS2 tag. MCF-7 cells were co-transfected with either *lincNORS*-MS2 vector or the empty vector and the pMS2-GST vector. 48 h after transfection, cells were lysed and proteins were quantified. Cell lysate containing 2 mg of protein was incubated with 60 μL Glutathione sepharose beads (GE Healthcare) for 3 h at 4 °C, followed by five washes with the lysis buffer to remove nonspecific bound proteins. Beads were further treated with 20 units of RNase-free DNase (QIAGEN) followed by two washes with NT2 buffer. For western blot analysis, beads were resuspended in SDS loading buffer, heated up to 95 °C for 5 min, and centrifuged for 1 min at 14,000 × *g*. The supernatant was loaded on an SDS-PAGE gel. For RNA analysis, 700 μL QIAzol lysis buffer were added onto the beads, and RNA was extracted as previously described. For liquid chromatography-mass spectrometry (LC–MS) analysis, proteins were recovered from beads and analyzed with the LTQ Velos Pro ion trap mass spectrometer. RAW files were searched using Proteome Discoverer 1.4 with SEQUEST HT.

**Ribonucleoprotein immunoprecipitation (RIP) analysis**. RIP analysis was performed following the protocol outlined by Abcam in Ivan lab. Briefly, nuclear extracts (native, no crosslink) from MCF-7 cells were prepared in RIPA buffer and 5% of the lysate was preserved as input RNA control. RALY, TRIM28 antibodies or IgG were added to the lysate and incubated overnight at 4 °C with gentle rotation. Complexes were captured using Dynabead Protein A (Life Technologies #10001D) and RNA was isolated using RNeasy kit (Qiagen). RNA immunoprecipitation performed by Macchi lab followed the protocol established by Keene et al., with some modifications[78]. Briefly, $2 \times 10^7$ MCF7 cells were lysed in lysis buffer (10 mM HEPES pH 7.4, 100 mM KCl, 5 mM $MgCl_2$, 0.5% NP-40, 1 mM DTT plus RNase and proteinase inhibitors) for 3 h at −80 °C and centrifuged at 12,000 × *g* for 30 min at 4 °C. The supernatants were collected and 1% of each sample was set aside as input while the remaining was incubated for 4 h at 4 °C with protein A magnetic beads (Invitrogen) coated either with 3 μg of an anti-RALY antibody (Bethyl, A302-069A) or with 3 μg of Rabbit IgG (Millipore). The beads were then washed four times with NT2 buffer (50 mM Tris–HCl pH 7.5, 150 mM NaCl, 1 mM $MgCl_2$, 0.05% NP-40, 1% urea) and RNA was isolated using TRIzol (Invitrogen). Purified RNA was then converted to cDNA using the RevertAid First Strand cDNA Synthesis Kit (Thermo Scientific), following the manufacturer's instructions. The qPCRBIO SyGreen Mix (PCR Biosystems) was used as a master mix for qRT-PCR. The reactions were performed on a CFX384 thermal cycler (Biorad) in technical duplicates for each target and for a total of four independent biological replicas. RALY RIP-seq data were previously published by Macchi lab and re-analyzed for lncRNA and cholesterol gene enrichment using the same method as previously described[44].

**Luciferase assay**. Oligos of *lincNORS* promoter region containing rs246185 and rs181766 reference or alternative allele were ordered as gBlocks fragments from IDT and inserted into pGL4 vectors. MCF7 were seeded at 70–80% confluence in 24-well tissue culture plates 24 h before transfection. Luciferase construct (1 μg) and pSV-β-galactosidase vector (0.1 μg), or their corresponding empty vector controls were transfected into cells according to manufacturer's protocol using Superfect reagent (Qiagen) with a DNA/Superfect ratio of 1:2.5 (wt/wt). The promoter activity was calculated from the ratio of firefly luciferase to β-galactosidase levels and expressed as arbitrary units.

**Colocalization analysis**. Colocalization analysis was performed to calculate the likelihood that the same causal SNP drives both *lincNORS* expression in human artery and skeletal muscle as well as the age of menarche GWAS signal at this locus. We queried the GTEx online server (https://www.gtexportal.org) to identify the tissues in which rs181766 is an eQTL for *lincNORS*. For each tissue with a *lincNORS* eQTL at rs181766, we used a 500 kb window (±250 kb) around rs181766 to test whether the eQTL signal colocalized with the age of menarche GWAS signal at this locus from Day et al.[53]. Colocalization analysis was performed in R (version 3.4.2) using the "coloc" package[52]. Data are presented as the posterior probability of the two signals colocalizing at this locus (PP4) and the posterior probability of the two signals colocalizing conditioned on there being a signal from both the eQTL and the GWAS data (PP4/PP3 + PP4). A higher value indicates a higher likelihood that the signal is shared.

**Statistical analysis**. Statistical analysis was performed in Excel, R (version 3.0.1, http://www.rproject.org/) or GraphPad Prism (version 6). Statistical significance of differences was determined by unpaired two-sided Student's *t*-test or ANOVA with post hoc comparison where appropriate unless stated otherwise in the figure legend. The difference was considered statistically significant when *p*-value was <0.05. Data are presented as mean ± SD, unless stated otherwise.

**Reporting summary**. Further information on research design is available in the Nature Research Reporting Summary linked to this article.

## Data availability

RNA-seq and BrU-seq data that support the findings of this study have been deposited in the National Center for Biotechnology Information Gene Expression Omnibus (GEO) and are accessible through the GEO Series accession numbers GSE153293 and GSE152799, respectively. All other relevant data are available from the corresponding author upon reasonable request. Source data are provided with this paper.

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

## Acknowledgements

We thank members of the IU Simon Cancer Center In Vivo therapeutics (IVT) Core for assistance in animal experiments. We also thank Dr. Je-Hyun Yoon for the technical advice on the MS2 RNA pulldown system, and Dr. Mervin Yoder for the critical comments and suggestions for the manuscript. This work was supported in part by NIH R01CA155332, Pancreatic Cancer Signature Center at Indiana University, and Biomedical Research Grant at Indiana University. M.G. was supported by the NIA IRP, NIH. C.M.N. was supported by the Foundation for Cellular and Molecular Medicine. D.L.C. was supported by 1K99HD099330-01. J.R. is supported by Genome Canada Genome Technology Platform Grant 12505, Canada Foundation for Innovation Project 33408.

## Author contributions

X.W. and M.I. conceived and designed the experiments. X.W., C.M.N., A.R., E.B., M.K.W., O.M.T., D.N.P., B.M., A.M. (McIntyre), A.B., M.B.P., S.M., and H.K. contributed to the

in vitro experiments. A.S.J. performed LC–MS analysis on total cholesterol content and metabolites ratio measurement. D.L.C. and W.P.B. performed an analysis of *lincNORS*-associated SNPs. X.Z., F.F., and K.P.N. generated and analyzed LTED-MCF7. S.M. generated samples from FTE cells. M.L.F. generated samples from patient-derived cell lines. M.G. contributed to the RNA pulldown experiment and A.M. (Mosley) performed the mass spectrometry analysis. A.R., T.T., and P.M. independently validated *lincNORS* interaction with RALY. The mouse experiments were performed by X.W. with assistance from IU Simon Cancer Center In Vivo therapeutics (IVT) Core and the tumor metastasis was analyzed by G.E.S. B.A.H., M.R., and K.P.N. performed the AmpliSeq and RNA-seq. M.T.P. and M.L. performed the sequencing and data analysis of BrU-seq. X.W., A.B. (Buechlein), C.I., J.R., and S.O. contributed to the bioinformatic analysis. X.W. prepared the figures for the manuscript. X.W., C.M.N., M.B.P., and M.I. drafted and edited the manuscript with input from all authors.

## Competing interests

The authors declare no competing interests.

## Additional information

Xue Wu[1,2], Cristina M. Niculite[2,3], Mihai Bogdan Preda[4], Annalisa Rossi[5], Toma Tebaldi[6,7], Elena Butoi[4], Mattie K. White[2], Oana M. Tudoran[8], Daniela N. Petrusca[2], Amber S. Jannasch[9], William P. Bone[10], Xingyue Zong[11], Fang Fang[11], Alexandrina Burlacu[4], Michelle T. Paulsen[12], Brad A. Hancock[13], George E. Sandusky[14], Sumegha Mitra[15,16], Melissa L. Fishel [15,17], Aaron Buechlein[18], Cristina Ivan [19], Spyros Oikonomopoulos[20], Myriam Gorospe[21], Amber Mosley [22], Milan Radovich[12,15], Utpal P. Davé [1,2,15], Jiannis Ragoussis [20], Kenneth P. Nephew [11,15,23], Bernard Mari [24], Alan McIntyre [25], Heiko Konig[2,15], Mats Ljungman[12,26], Diana L. Cousminer[27], Paolo Macchi[5] & Mircea Ivan [1,2,15]✉

[1]Department of Microbiology and Immunology, Indiana University School of Medicine, Indianapolis, IN 46202, USA. [2]Department of Medicine, Indiana University School of Medicine, Indianapolis, IN 46202, USA. [3]"Victor Babes" National Institute of Pathology, Bucharest, Romania. [4]Institute of Cellular Biology and Pathology "Nicolae Simionescu", Bucharest, Romania. [5]Laboratory of Molecular and Cellular Neurobiology, Department of Cellular, Computational and Integrative Biology - CIBIO, University of Trento, Trento, Italy. [6]Laboratory of Translational Genomics, Department of Cellular, Computational and Integrative Biology - CIBIO, University of Trento, Trento, Italy. [7]Yale Cancer Center, Yale University School of Medicine, New Haven, CT 06520, USA. [8]The Oncology Institute "Prof Dr. Ion Chiricuta", Cluj-Napoca, Romania. [9]Metabolite Profiling Facility, Bindley Bioscience Center, Purdue University, West Lafayette, IN 47907, USA. [10]Department of Genetics, Department of Systems Pharmacology and Translational Therapeutics, Institute of Translational Medicine and Therapeutics, Perelman School of Medicine, University of Pennsylvania, Philadelphia, PA 19104, USA. [11]Department of Cellular and Integrative Physiology, Indiana University School of Medicine, Indianapolis, IN, USA. [12]Departments of Radiation Oncology and Environmental Health Sciences, Center for RNA Biomedicine, University of Michigan, Ann Arbor, MI 48109, USA. [13]Department of Surgery, Indiana University School of Medicine, Indianapolis, IN 46202, USA. [14]Department of Pathology and Laboratory Medicine, Indiana University School of Medicine, Indianapolis, IN 46202, USA. [15]Melvin and Bren Simon Cancer Center, Indiana University, Indianapolis, IN, USA. [16]Department of Obstetrics and Gynecology, Indiana University School of Medicine, Indianapolis, IN 46202, USA. [17]Department of Pharmacology and Toxicology, Department of Pediatrics, Wells Center for Pediatric Research, Indiana University School of Medicine, Indianapolis, IN 46202, USA. [18]Indiana University Center for Genomics and Bioinformatics, Bloomington, IN 47405, USA. [19]Center for RNA Interference and Non-coding RNAs, The University of Texas MD Anderson Cancer Center, Houston, TX 77030, USA. [20]Department of Human Genetics, McGill University and Genome Quebec Innovation Centre, McGill University, Montréal, QC, Canada. [21]Laboratory of Genetics and Genomics, National Institute on Aging, National Institutes of Health, Baltimore, MD 21224, USA. [22]Department of Biochemistry and Molecular Biology, Indiana University School of Medicine, Indianapolis, IN 46202, USA. [23]Medical Sciences, Indiana University School of Medicine, Bloomington, IN, USA. [24]CNRS, IPMC, FHU-OncoAge, Université Côte d'Azur, Valbonne, France. [25]Rogel Cancer Center, University of Michigan, Ann Arbor, MI 48109, USA. [26]Centre for Cancer Sciences, Biodiscovery Institute, Nottingham University, Nottingham, UK. [27]Division of Human Genetics, Department of Pediatrics, The Children's Hospital of Philadelphia, Perelman School of Medicine, University of Pennsylvania, Philadelphia, PA 19104, USA. ✉email: mivan@iu.edu

