## [Peer Review File · Nature Communications]

Reviewers' comments, first round:

Reviewer #1 (Remarks to the Author):

NORS: A lincRNA Regulator of Sterol Biosynthesis"

In this article, Wu and colleagues investigate the functions of a lincRNA which they name NORS (Noncoding Oxygen-Sensitive Regulator of Sterol Biogenesis) transcribed from the MIR193BHG locus on chr16. NORS is initially identified from a RNAseq-based screening for novel transcripts modulated by oxygen tension in the breast cancer cell line MCF-7, (ENSG00000262454/MIR193BHG). The lincRNA transcript produced from this locus, NORS, is elevated also in two non-transformed cell lines at 0.2% O₂ and other cancer cell lines at different hypoxia rates: ER alpha positive BC and TNBC at both 0.2% and 2% O₂; lung, colorectal and ovarian cancer cell lines at 0.2% O₂ and patient-derived pancreatic cancer cell lines and CAF at 1% O₂. Low oxygen transcriptional response is known to be mediated by Hypoxia Inducible Factors (HIFs). Using meta-analysis of genome-wide mapping of HIF-binding sites they hypothesize that a regulatory region within the LINC02130 locus contains the key regulatory region rather than the proximal promoter itself. Meta-analysis of ChIP-PET sequencing of RNAPII-immunoprecipitated complexes in MCF-7 cells suggests potential interactions between NORS proximal promoter and the LINC02130 region. By knocking down HIF-1 α , HIF-2 α and FOSL2 in MCF-7 and MDA-MB-231 cells they showed that hypoxic regulation of NORS expression is mediated by HIF-2 α and FOSL2. Using cancer gene expression datasets, they highlight a correlation of NORS with hypoxic markers, particularly in TNBC and PAAD, and additional pathways such as glucose and lipid metabolism and estrogen response.

First, the authors test a connection between NORS and glucose metabolism by comparing breast cancer cells in low/high glucose showing a difference in the NORS levels between normoxic and hypoxic conditions at 1% O₂ only in the high glucose conditions. Secondly, they investigate if NORS is controlled by estrogen/ESR1. According to ChIP-Atlas, there are multiple ESR1 binding signals detected in several breast cancer cell lines and the majority of these binding sites showed increased chromatin opening (detected by DNase HS) in response to estradiol. Furthermore, ChIP-seq data from ENCODE showed that ESR1 pioneer factors and functional partners including FOXA1, GATA3, and NCOA3/SRC3 were recruited to the NORS-LINC02130 region. Supporting the sterol receptor connection, NORS has been previously reported by other authors among the transcripts significantly downregulated in endocrine therapies-resistant, long-term estrogen-deprived MCF-7 cells. From TCGA database, NORS also exhibits higher expression in ER/PR positive breast cancers. Using tissue expression data from GTEx, they expand their analysis to normal tissues. Despite the discrepancy between datasets, the authors highlight that NORS appear to be more abundant in steroid hormone relevant tissues (breast, adipose tissue, prostate, adrenal gland, testis, and skeletal muscle).

Thirdly, to investigate the correlation between NORS and lipid metabolism, they started with RNAseq profiling of breast cancer cell lines that showed an upregulation of cholesterol biosynthetic pathway following NORS inactivation, and validated it with qPCR analysis. They generated sh, siRNA and overexpression constructs and then used them to study the effect of hypoxia 1% O₂ on three breast cancer cells. Silencing led to upregulation of sterol biosynthesis transcripts under hypoxia, in particular HMGCR, HMGCS1, MVD, SQLE, LSS and MSMO1 mRNAs. Two of these enzymes downstream of SREBPs activation were also upregulated in one prostate cancer cell line. Also, using an enzymatic-based assay for MCF7 and MDA-MB-231 and liquid-chromatography-mass spectrometry for MDA-MB-231 only, they show that total cholesterol increases after NORS downregulation, while its overexpression had a suppressive effect on the lipid content. NORS appears to regulate other branches of sterol metabolism in particular, STARD4, involved in intracellular sterol transport, and SCD in both MCF7 and PC-3. The authors conclude that NORS might negatively regulate sterol biosynthesis programs in a broad context, regardless of the gender or sex hormone receptor status.

Upregulation of lipid biosynthesis pathway has been previously shown to promote resistance to hormonal therapy, stimulate the metastasis and decrease overall survival in diverse cancer settings. To this end, the authors examined the effects of NORS excess or depletion in standard assays for cell viability and invasion under different environmental conditions. Knockdown of NORS in TNBC cells had a promoting effect on invasion, and such impact was blunted by simvastatin treatment. Similar but more modest effects of NORS were observed in cell viability assays and statin treatment in MCF7 and MDA-MB-231. In MCF-7 cells, the presence of metformin and microenvironment stress (1%O₂ and low glucose) magnified the suppressive effect of NORS overexpression on cell viability. Tumor xenograft were generated using MDA-MB-231 stably expressing full-length NORS, the NORS shRNA construct, or the corresponding controls. Briefly, while NORS knockdown had no significant impact on primary tumor growth, it significantly accelerated the metastatic process, whereas NORS overexpression had the opposite effect. They investigated the mechanistic action of NORS on the lipid biosynthesis studying the nascent transcript rate in NORS-depleted MCF-7 cells, the results suggested that NORS functions as a repressor of transcription. MS2-tagged NORS RNA pulldown from MCF7 cells, did not detect the presence of nuclear receptors directly or indirectly affecting sterol biogenesis, the NORS "interactome" was enriched in histones instead. Proteins mostly studied for roles in mRNA maturation, in particular RALY, interacted endogenously with NORS and knockdown of RALY abolished the effects of NORS on steroid biosynthesis program. Finally, the authors interrogates GWAS and QTLs datasets. In the case of NORS, there are five SNPs and they are among the top SNPs associated with primary or secondary steroid-related phenotypes including puberty timing and measurement, androgenetic alopecia, and adult stature.

Overall the manuscript attempts with partial success to characterize a lincRNA and study its function mainly in a cancer setting. The study is interesting because it identifies a modulator of the sterol biosynthesis in cancer biology, but we fear that there are some major conceptual issues due to a rather erratic plan of investigation that does not end up answering what we think are the three critical questions:

What does NORS do? How is NORS regulated? What's the impact on normal tissue development?

Considering that the third point has not been properly addressed and more extensive experiments need to be done in order to clarify the supposed connection between this lincRNA and the derived phenotypes (from GTEX), we recommend that the authors to consider this third point for further development (and a separate manuscript). Indeed, this reviewer thinks the overarching aim of this manuscript is too ambitious, i.e. characterize NORS at a physiologic and pathologic level. Although very interesting, the connection of NORS with physiological processes is supported only by preliminary data. It might be worth to validate the analysis of available datasets done in fig 7 with experiments aimed at characterizing regulatory elements, engineering or reproducing SNPs, using even mouse models of steroid-related phenotypes like timing of menarche and genital development. In the present state, these meta-analyses are too preliminary and too correlative to support any conclusion. It is this reviewer opinion that characterization of NORS should be focused more on the cancer setting (as the totality of experiments uses almost exclusively cancer cell lines) rather than try to speculate about a possible physiological role.

This consideration brings to another important note that more characterization of the link with the estrogen response/nuclear receptor should be more explicit. Indeed, the authors' deduction that "there seems to be a clear connection between NORS locus and estrogen/ESR1" and their knowledge that "NORS has been reported among the transcripts significantly downregulated in endocrine-therapy resistant, long-term estrogen deprived MCF7 cells. NORS also exhibits higher expression in ER/PR positive breast cancers" should be enough to further investigate this key point.

Of three major connection found (NORS and glucose metabolism, estrogen response and lipid metabolism) only the lipid metabolism has been investigated more in depth. Considering that all cells were cultured in high glucose media, whether the consensus at least for the breast cancer cell lines, is to culture them in 1 g/L glucose (normal glycemia in humans is 0.7-1.2 g/L), one cannot exclude that the real driving force of the NORS expression is high glucose itself. To exclude that glucose is a necessary and sufficient condition to activate NORS sterol biosynthesis regulation, all experiments in fig 1 should be done in low glucose conditions as well.

Although the oxidative-phosphorylation pathway is one of the most upregulated in the gene set enrichment analysis in both cancer cells and normal tissues analysis (fig 3c and 3e), it is not mentioned in the manuscript nor it is investigated. Only one experiment (fig S8) was done to test the effect of Metformin (and only in the MCF7), using low and high glucose and 1% O₂ conditions. Unfortunately, there is no follow up or any mention to a possible explanation of these effects so this reviewer suggests to either expand this point or edit the figure.

Furthermore, the reviewer thinks that more consistency should be used for the oxygen deprivation conditions. The variability between the experiments and the cell lines used ranges from 0.2% to 2% O₂. According to what stated by the authors in the text "While HIF-1 α is generally thought to mediate acute adaptation to severe hypoxia, HIF-2 α appears predominantly involved in the chronic responses to milder, often physiologically relevant, oxygen deprivation. Specifically, HIF-2 α exerts context-dependent positive or negative effects on cell growth and functions as a driver of stemness in normal or neoplastic tissues [15, 16].", and considered their results point at HIF-2 α only regulating NORS's hypoxic response, I suggest that a milder oxygen deprivation should be used (2%O₂) across the study.

Regarding the connection of NORS with the lipid metabolism, the results show an upregulation of the downstream enzymes however, considered the extensive literature regarding the role of the master regulators of the sterol biosynthesis SREBPs, it would be essential to understand if NORS collaborates with SREBPs for example by seeing if these TFs localize in the nucleus under hypoxic conditions (fig 1 g).

This reviewer found very interesting the relationship between NORS and RALY. To further confirm this mechanistic link, it would be nice if the authors could investigate how NORS regulates sterol biosynthesis through RALY interaction (Does RALY interact with the gene's promoters, RALY ChIP?).

Overall, this reviewer would be interested in re-considering a majorly revised manuscript. It is noted that the premised and initial findings are of interest but there is a strong need to consolidate the molecular findings without the need of major physiological extrapolations.

Minor comments:

Introduction:

"The outcome is a rare case, to our knowledge, of an informative convergence between the experimentally-determined function of a lincRNA and the phenotypic associations of the corresponding locus." Overstated, as there are not conclusive data on phenotypic association (the manuscript discuss invasion but then highlights alopecia?).

Results :

Fig. 1: (e) the method used for the experiment done in patient-derived cell lines and CAFs should be consistent with the other cell lines (qPCR).

Legends (c-e) not correct.

RNA scope, where are the negative and positive controls?

Fig. 3b: 84 TNBC and 41 PAAD tumor samples are used, there is no mention in the methods, where they come from? Are they primary, metastatic?

The paragraph title "NORS Modulates Sterol Biosynthesis" should probably be modified into something more consistent with the paragraph content. Indeed, the experiments described here are aimed at studying the effects of NORS on sterol biosynthesis in cancer and on invasion and tumour growth.

Methods:

RNAscope, missing information (kit and products catalog number, probes, dilution and channels used)

WB, missing information about concentration of antibodies used

Xenograft models, how many animals used?

Discussion:

It needs to be revised according to the comments, reconsidering all the discussion done about the puberty onset and physiology setting that covers more than half of the total space dedicated to this section. The importance given to this aspect of the paper is not justified by any of the experiments or the experimental design of the study.

References:

written twice, of which the first one of 69 is the incorrect one.

Regarding the upregulation of the sterol biosynthesis pathway promoting resistance to hormonal therapy in cancer, this reviewer suggests adding the following relevant citations:

Nguyen, V. T. et al. Differential epigenetic reprogramming in response to specific endocrine therapies promotes cholesterol biosynthesis and cellular invasion. *Nature Communications* 10044 6 ,

Nelson ER1, Wardell SE, Jasper JS, Park S, Suchindran S, Howe MK, Carver NJ, Pillai RV, Sullivan PM, Sondhi V, Umetani M, Geradts J, McDonnell DP. *Science*. 2013 Nov 29;342(6162):1094-8. doi: 10.1126/science.1241908. 27-Hydroxycholesterol links hypercholesterolemia and breast cancer pathophysiology.

Reviewer #2 (Remarks to the Author):

The manuscript entitled "NORS: a LincRNA Regulator of Sterol Biosynthesis" by Wu et al is announced as an original work contribution. In this work the authors identify a long-coding RNA which they name NORS by screening for dynamically regulated transcripts in response to oxygen tension in a human cancer cell line. Using co-expression network analysis from publically available datasets and lncRNA perturbations studies mostly done in vitro the authors assert that NORS plays a role in sterol biosynthesis.

Overall the work is novel, timely and provides interesting insights on the contributions of lncRNAs in human metabolic regulatory networks. This is particularly noteworthy since extrapolation of murine effects to human disease is often a challenge in the field of lncRNA biology owing to their poor conservation. Nevertheless, the manuscript falls short on multiple fronts including a number of technical and methodologic concerns as well as lack of detailed mechanistic insight.

1. The work almost exclusively uses RNAi for functional studies. siRNAs and shRNA require machinery and are known to have substantial off target effects that notoriously influences sterol regulatory pathways. In addition, their sole use is not considered acceptable for thorough interrogation of lncRNA functions. If the authors wish to make a claim about regulation of sterol biosynthesis by NORS, genetic inactivation tools such as CRISPR are absolutely required and can be readily accomplished nowadays. What is the effect of knockout of NORS on sterol biosynthetic genes and does unbiased gene expression in this context reveal other pathways? Does overexpression of NORS in this context rescue the phenotype?

2. Although the manuscript focuses on a human specific lncRNA which can be difficult to perturb, very little evidence is provided to substantiate proposed mechanisms as operational in vivo. This is particularly important since cellular models have significant limitations in studying sterol regulation and the phenotype reported here is very subtle. Since the authors are invoking a trans mechanisms of action what are the consequences of NORS overexpression in mouse liver (using adeno or AAV)? Do the authors observe changes in liver and serum cholesterol levels as well as gene expression of SREBP and target genes? In addition, the manuscript would be significantly bolstered if the authors can use the same approach in a humanized mouse liver model. These studies will provide important information on the functional conservation of this lncRNA which can be preserved despite sequence evolution as well as help tie the main conclusions of the paper with a more relevant in vivo model.

3. The evidence that the lncRNA and microRNAs are independently regulated is not sufficient and based mostly on qPCR data in figure 1. Are the authors proposing that the expression of the microRNA does not require the production of the host gene lncRNA? Figure 2A does not support that notion since it shows that the induction in newly synthesized RNA with oxygen tension includes the intronic regions where the microRNAs are embedded. The authors should disrupt microRNA production using CRISPR and show that effects of NORS perturbation are still preserved.
4. Very little information is provided on the characterization of the NORS transcript itself. What is the copy number of NORS in MCF-7 cells? The single molecule FISH supports a very low copy number. In addition, the authors should show evidence of in vitro protein transcription/translation using a positive control.
5. The figures are very poorly rendered in particular the genome browser shots make it impossible to decipher critical information about the relationship of the lncRNA transcript and embedded microRNAs and neighboring genes. The figures need to be of much higher resolution with font size and illustrations that make it very clear where the lncRNAs transcripts are produced and their relationship to other transcripts.
6. The authors should present unbiased analysis of the BrU-seq in the supplement.
7. Please describe the source data in the methods and more detailed methodology including how gene expression is normalized and detailed primer sequences. Please be sure to report error bars for all graphs for example are the results of the first panel in 2C significant?

Reviewer #3 (Remarks to the Author):

NCOMMS-18-25313-T

"NORS: A lincRNA Regulator of Sterol Biosynthesis"

Remarks to the Authors:

Xue Wu et al. provide a profound functional analysis of the lncRNA NORS.

The authors show that NORS is hypoxia inducible, mediated by HIF-2 and FOSL2 transcription factors. By numbers of experimental and bioinformatics data, they characterized NORS genomic location and environment as well as NORS regulation in hypoxia and in diverse cancer types, including patient-derived carcinoma cells. In vivo, NORS correlation analysis from tumor and normal tissues data revealed that NORS contributes to a hypoxic/ tumorous network. By gene set enrichment analysis (GSEA) of NORS-correlated genes, the authors identified pathways correlated with NORS beyond the hypoxic response. NORS involvement in estrogen response and especially the cholesterol biosynthesis pathway has been further investigated in more detail. It is demonstrated that NORS interaction with RALY is crucial for its regulatory role in sterol synthesis. These findings are linked to NORS ability to inhibit cell invasion and metastasis of breast cancer cells.

The results are novel and of outstanding interest to others in the community and in the wider field. The authors characterized lncRNA NORS in depth, including up-stream factors and down-stream pathways. For instance, the data provide evidence for a new layer in the complexity of the hypoxia/ sterol biosynthesis network up to the point of NORS being a new tumor suppressor. The methods used are appropriate, state of the art and well described. The claims have been adequately discussed.

The data are convincing and I do not see the need for further experiments. In contrast, at some points I suggest to remove data to improve clarity, as outlined below. Some minor issues should be addressed:

Summary:

The summary does not reflect the magnificent findings of the study.

Results section, "Identification and Characterization of NORS":

The usage of different oxygen levels (0.02 %, 1 %, 2 %) appears somewhat confusing. The statement "...was also detected at 2% O₂, a more physiologically relevant oxygen concentration" is not correct. We all know that oxygen tension is highly diverse in different tissues, even under physiological conditions. I suggest removing Figure 1f, or, alternatively, showing an oxygen response kinetic (0.02 up to 2 % O₂) of NORS levels for cell lines MCF-7 and MDA-MB-231 as Figure 1f. The latter option would well fit to the Discussion section.

Please check the link to Table S2 in "...confirmatory evidence for NORS's oxygen sensitivity (Table S2)." Do you mean Table S1?

Figure 1: Please indicate time of hypoxia in legend of Figure 1b. The attribution of Figure 1c-e in the legend is wrong ("b" is c, "c" is d, "d" is e). Please indicate that the same cycle number was used in (e).

Figure S1: Please provide higher resolution pictures for S1a-d.

Results section, "Hypoxic Regulation of NORS":

Please improve resolution of Figure 2a, e.

In Figure 2b it is shown that NORS is unaffected by DMOG in MCF-7 cells. A 2-OG analog that inhibits prolyl-hydroxylases will activate all HIF isoforms, including HIF-2. In the same cell line, NORS is elevated by 1 % oxygen. Western blots in Figure 2c suggest that HIF-2alpha is present in MCF-7 cells and, thus, should influence NORS level similarly as in MDA-MB-231 cells. Does DMOG not induce HIF-2alpha in MCF-7 cells? Showing Western blots for HIF-2alpha in DMOG treated cells (Figure 2b) and comparing HIF-2alpha levels in MCF-7 and MDA-MB-231 cells is recommended.

Figure 2c: In MDA-MB-231 cells the LaminB1 signal is strongly overexposed, leading to a burn out of chemiluminescence substrate that in turn causes white bands within the black signals. Such overexposed signals are insufficient for normalization. I suggest to provide LaminB1 signals of lower intensity.

Figure 2d: Please provide significance levels for the comparison of 21 % vs. 0.2 % O₂ in VHL+ and VHL- cells.

Results section, "NORS Regulatory Signals In Vivo: Insights from Normal and Diseased Tissues":

Authors should consider to rephrase the sentence: "...we observed that the response of NORS to decreased oxygen concentration is restricted by glucose availability, supporting a connection between NORS and glucose metabolism (Figure 3d)." Keeping in mind that during hypoxia, glucose represents the only substrate in energy metabolism (ATP production by glycolysis only), glucose deprivation in hypoxia may cause a drop in general ATP availability that also affects overall transcription rate. Furthermore, a correlation of NORS with glycolytic factors as shown in Figure 3c is expected, as all are hypoxia inducible. For me, these data do not satisfactory show a direct connection between NORS and glucose metabolism as suggested by the authors. However, this is not the focus of this study; thus, authors may consider investigating the NORS-glucose metabolism axis separately.

Results section, "Mechanistic insights into NORS action on the steroid biosynthetic program":

Please provide a short description regarding TRIM28. Is the binding of NORS to TRIM28 at 1 % oxygen significant (Figure 6c)?

Following the authors argumentation, significance levels for si_CTRL vs. siRALY in cells treated with si_NORS_s during hypoxia and normoxia should be indicated (dark green columns in Figure 6f). For clarity, I suggest not to show data for si_NORS_w in Figure 6f, as it does not contribute to

the conclusion.

Significance levels are also missing in Figure 6b.

Discussion, para 4:

I suggest to rephrase "physiological hypoxia", as hypoxia represents a condition when oxygen demand exceeds supply. "physiological low pO₂" would be more appropriate.

Methods, "Subcellular fractionation":

Please mention the conditions of centrifugation to obtain the cytoplasmic fraction (10,000 x g?).

Methods, "RNA extraction and qPCR analysis":

The link to Table S2 for the list of gene-specific primers is wrong. Supplementary Table "oligos" would fit.

Michael Föhling

Reviewer #4 (Remarks to the Author):

The authors report the first characterization of a noncoding RNA named NORS. They showed that NORS-regulation involved growth signaling and that NORS repressed steroid biosynthesis (including sterol) transcriptional programs. They used genome-wide association studies to support the biological relevance of NORS with the identification SNPs on enzymes from the sterol pathway overlapping NORS's regulatory region. Finally, they report the identification of an interaction of NORS with the RALY protein, which was shown to repress cholesterol biosynthetic genes expression in mouse liver.

Although this study open new perspectives in the identification of possible new actors involved in the regulation of steroid/sterol metabolism, it is mainly correlative and thus preliminary. Importantly, this study lacks experimental evidence that NORS controls cholesterol neo-synthesis.

Reviewer #5 (Remarks to the Author):

I think Wu et al make an interesting observation, which is the regulation of NORS levels under different conditions. Unfortunately, the quality (resolution) of many of the figures is pretty low and it is often difficult to read/evaluate. My review will focus mostly on the proteomic analysis of NORS, however, I would like to highlight that authors should improve the quality of the figures by using images with higher resolution.

PROTEOMIC ANALYSIS OF NORS

I find some limitations in the NORS 'interactome analysis' that require discussion/clarification or/and experimental validation. It is not the same to identify proteins that interact with NORS and to establish the NORS interactome. The first refers to cherry pick proteins from the proteomics through a hypothesis driven approach aiming at finding NORS regulators and understanding that the dataset is corrupted by false positives. The second refers to the comprehensive determination of the repertoire of proteins that interact with NORS, minimizing false positives to provide a resource for the community. Each approach have different requirements. While the first would focus on the downstream experimental validation of the protein's function (which I don't think is enough here), the second will require extensive validation of the capture as well as stringent analysis of the data. As it is, I don't think this work fulfill the requirements to call for a 'NORS interactome'.

1. First, NORS is expressed by transitory transfection, which means that NORS levels are expected

to be non-physiological. Experiments determining endogenous and ectopically expressed NORS and how they compare are required to balance the conclusions.

2. As the capture relies on MS2, native purification conditions are used. This implies that authors purify protein that bind directly and indirectly to NORS and this should be stated in the text.

3. It is known that MS2 capture when using in cellulose approaches are very noisy due to the purification of indirect binders and unwanted RNAs. To convince me about this proteomic analysis, authors should exhaustively analyze the RNA pool isolated by MS2 capture in their experimental conditions. This can be done through RNAseq (optimally) or bioanalyser (at least to show that the dominant RNA population has the size of NORS). Enrichment over ribosomal RNA only tells me the degree of enrichment against that particular (highly abundant) RNA, but does not provide any information about the specificity of the capture beyond that. Just looking at the numbers, knowing the abundance of rRNA and NORS (FISH analyses) per cell, rRNA would still be dominant in the pull down in spite of the enrichment showed in Figure 6b (30 fold). This would explain why authors identify basically the whole ribosome in the pull down. Would this mean that the lincRNA is translated or just relate to contaminant RNA? As a positive aspect, I understand from the figure (not clear in the figure legend) that the control is absence of an RNA with MS2. This is good as the contaminant RNA will mostly come from MS2-bp unspecific binding when present in excess. I suggest to the authors to clarify this point and describe in more detail the experimental approach and the controls used.

4. To convince me/us of the high stringency of their method, they should show the proteins purified by the MS2 capture in experimental and controls by a general protein staining, in particular high sensitivity silver staining. They should detect an enrichment of specific bands in the specific capture over the controls.

5. Figure 6b shows that enrichment of candidate RBPs in the experimental over control is residual (see RALY and TRIM28). This is unconvincing to me as I would expect a black (negative control) and white (experimental) situation if the purification has enough stringency. Can authors comment on this?

6. SAINT employs protein length to normalize/correct the spectral counts and this affects probability, which is good as the bigger of protein is, more chances of getting spectral counts from different peptides are. However, authors employ here a very loose SAINT cut off (0.5). Choi et al., Nat Meth 2011, defined 0.8-0.9 SAINT values as stringent cut offs equivalent to 5% and 2% FDR, respectively. From my point of view, a cut off of 0.5 is just not acceptable in a noisy experiment such as specific RNA capture by MS2. Note that most of the proteins in the 0.5 statistical threshold display very small fold change over controls (that is supposed to lack the RNA of interest). Under the limitations of specific RNA capture (low signal over noise, contamination with unwanted RNAs, use of native conditions, etc), one should apply stringent statistical standards to minimize false positives. I assume that there is not currently a perfect method to capture single RNA species, but it is important to keep the highest standards to avoid delivery of useless data. It is important that authors include a legend or a more detailed headings (e.g. what is fold change A and B) in the supplementary table, so it is easily accessible.

7. One of the problems is that RALY, the proteins that authors focus afterwards on, has 0.5 SAINT value (2.39 and 1.63 fold change; which seems low to me when comparing to a negative control). Based on this value, there is not strong support from the proteomic data on the NORS/RALY interaction. This raises questions about why authors selected this protein. This needs clarification as for me, other proteins in the list would represent much better candidates for follow up studies (e.g. TRIM28).

8. If authors wants to say that the data provided is the 'NORS interactome', they should validate multiple proteins falling under different SAINT scores and exhibiting different fold changes by an orthogonal approach.

9. Most of the isolated proteins appear to me as highly abundant (authors mentioned histones, but they also capture the near-complete ribosome). A density plot comparing protein or RNA abundance of the whole proteome and NORS proteome would help to rule out whether this approach just enriches for abundant proteins. This can be done with data already available in repositories. Moreover, as NORS is a (mostly) nuclear RNA, nuclear GO annotation should be prevalent over cytoplasmic in the 'NORS' interactome.

10. The validation by RIP is also unconvincing for me. RALY is a pretty good RBP in terms of UV crosslink-ability and I thus expect RALY produce quite extensive fold enrichment in RIP assays. However, authors only see 3-4 fold enrichment in normoxia and 7.5 in hypoxia – which correlate with higher NORS levels. RIP fold enrichment for TRIM28 is not convincing at all in spite of the

high SAINT score. Said that, I think this study would gain interest if authors would strength their point by elucidating whether RALY/TRIM28 binds NORS directly, and where these interactions occur. I suggest authors check first whether there is any RALY or TRIM28 iCLIP, HITS-CLIP, eCLIP dataset available in ENCODE to take advantage of readily available data. If not, performing this experiment with add value to the paper.

OTHER COMMENTS

11. Figure 1. Is abundance of the RNAs located around NORs increased upon hypoxia?
12. Figure S1C and D are not readable. So I cannot make any conclusion about them.
13. Figure 1h. To show reproducibility, it would help to provide a quantification of the RNAs dots detected in nucleus vs cytoplasm and in high and low oxygen from multiple biological replicates.
14. Figure 2c. Could you analyse non-responsive RNAs and cell death markers to rule out that HIF2alpha depletion affects the capacity of the cell to deal with hypoxia and the effects are thus indirect? Same applies to FOSL2 (panel f).
15. Page 6. 'NORS was supported by Coding Potential Assessment Tool (CPAT) analysis which yielded a very low coding potential score, comparable to those of MALAT1 and NEAT1 (Figure S1e).' Can you support this using any available ribosome profiling data?

We thank all reviewers for their genuine interest in our results. Based on their valuable suggestions, the revised manuscript presents a sharper portrait of a new genetic element with rich genetic associations. In summary, the hormonal (estrogen) connection is now developed as a major feature of this locus, in addition to hypoxia. Mechanistically, we show additional evidence that NORS predominantly affects the cholesterol biogenesis program at transcriptional rate level (rather than RNA stability) and we expanded the evidence for RALY as key mediator for NORS' activity. We are the first to admit that from a mechanistic standpoint our study is only intended as a first step towards understanding this previously overlooked genetic element. Further, we present experimental evidence for an interplay involving NORS, RALY and sterol biogenesis RNAs. We took to heart the valuable points raised by rev. 5 and generated a more rigorous justification for our focus on RALY focus. The revised manuscript also presents new analyses to formally connect this lincRNA to GWAS human phenotypes as well as the first functional analysis of corresponding snp variants (per rev.1's suggestion). We also stated more clearly that based on the body of evidence included in the manuscript, NORS is most likely relevant for the extrahepatic "facets" of sterol/steroid biology rather than liver cholesterol biosynthesis. Detailed replies to each reviewer follow below:

Reviewer #1

In this article, Wu and colleagues investigate the functions of a lincRNA which they name NORS (Noncoding Oxygen-Sensitive Regulator of Sterol Biogenesis) transcribed from the MIR193BHG locus on chr16. NORS is initially identified from a RNAseq-based screening for novel transcripts modulated by oxygen tension in the breast cancer cell line MCF-7, (ENSG00000262454/MIR193BHG). The lincRNA transcript produced from this locus, NORS, is elevated also in two non-transformed cell lines at 0.2%O₂ and other cancer cell lines at different hypoxia rates: ER alpha positive BC and TNBC at both 0.2% and 2% O₂; lung, colorectal and ovarian cancer cell lines at 0.2% O₂ and patient-derived pancreatic cancer cell lines and CAF at 1%O₂. Low oxygen transcriptional response is known to be mediated by Hypoxia Inducible Factors (HIFs). Using meta-analysis of genome-wide mapping of HIF-binding sites they hypothesize that a regulatory region within the LINC02130 locus contains the key regulatory region rather than the proximal promoter itself. Meta-analysis of ChIA-PET sequencing of RNAPII-immunoprecipitated complexes in MCF-7 cells suggests potential interactions between NORS proximal promoter and the LINC02130 region. By knocking down HIF-1 α , HIF-2 α and FOSL2 in MCF-7 and MDA-MB-231 cells they showed that hypoxic regulation of NORS expression is mediated by HIF-2 α and FOSL2. Using cancer gene expression datasets, they highlight a correlation of NORS with hypoxic markers, particularly in TNBC and PAAD, and additional pathways such as glucose and lipid metabolism and estrogen response. First, the authors test a connection between NORS and glucose metabolism by comparing breast cancer cells in low/high glucose showing a difference in the NORS levels between normoxic and hypoxic conditions at 1% O₂ only in the high glucose conditions. Secondly, they investigate if NORS is controlled by estrogen/ESR1. According to ChIP-Atlas, there are multiple ESR1 binding signals detected in several breast cancer cell lines and the majority of these binding sites showed increased chromatin opening (detected by DNase HS) in response to estradiol. Furthermore, ChIP-seq data from ENCODE showed that ESR1 pioneer factors and functional partners including FOXA1, GATA3, and NCOA3/SRC3 were recruited to the NORS-LINC02130 region. Supporting the sterol receptor connection, NORS has been previously reported by other authors among the transcripts significantly downregulated in endocrine therapies-resistant, long-term estrogen-deprived MCF-7 cells. From TCGA database, NORS also exhibits higher expression in ER/PR positive breast cancers. Using tissue

expression data from GTEx, they expand their analysis to normal tissues. Despite the discrepancy between datasets, the authors highlights that NORS appear to be more abundant in steroid hormone relevant tissues (breast, adipose tissue, prostate, adrenal gland, testis, and skeletal muscle).

Thirdly, to investigate the correlation between NORS and lipid metabolism, they started with RNAseq profiling of breast cancer cell lines that showed an upregulation of cholesterol biosynthetic pathway following NORS inactivation, and validated it with qPCR analysis. They generated sh, siRNA and overexpression constructs and then used them to study the effect of hypoxia 1%O₂ on three breast cancer cells. Silencing led to upregulation of sterol biosynthesis transcripts under hypoxia, in particular HMGCR, HMGCS1, MVD, SQLE, LSS and MSMO1 mRNAs. Two of these enzymes downstream of SREBPs activation were also upregulated in one prostate cancer cell line. Also, using an enzymatic-based assay for MCF7 and MDA-MB-231 and liquid-chromatography-mass spectrometry for MDA-MB-231 only, they show that total cholesterol increases after NORS downregulation, while its overexpression had a suppressive effect on the lipid content. NORS appears to regulate other branches of sterol metabolism in particular, STARD4, involved in intracellular sterol transport, and SCD in both MCF7 and PC-3. The authors conclude that that NORS might negatively regulate sterol biosynthesis programs in a broad context, regardless of the gender or sex hormone receptor status.

Upregulation of lipid biosynthesis pathway has been previously shown to promote resistance to hormonal therapy, stimulate the metastasis and decrease overall survival in diverse cancer settings. To this end, the authors examined the effects of NORS excess or depletion in standard assays for cell viability and invasion under different environmental conditions. Knockdown of NORS in TNBC cells had a promoting effect on invasion, and such impact was blunted by simvastatin treatment. Similar but more modest effects of NORS were observed in cell viability assays and statin treatment in MCF7 and MDA-MB-231. In MCF-7 cells, the presence of metformin and microenvironment stress (1%O₂ and low glucose) magnified the suppressive effect of NORS overexpression on cell viability. Tumor xenograft were generated using MDA-MB-231 stably expressing full-length NORS, the NORS shRNA construct, or the corresponding controls. Briefly, while NORS knockdown had no significant impact on primary tumor growth, it significantly accelerated the metastatic process, whereas NORS overexpression had the opposite effect. They investigated the mechanistic action of NORS on the lipid biosynthesis studying the nascent transcript rate in NORS-depleted MCF-7 cells, the results suggested that NORS functions as a repressor of transcription. MS2-tagged NORS RNA pulldown from MCF7 cells, did not detect the presence of nuclear receptors directly or indirectly affecting sterol biogenesis, the NORS “interactome” was enriched in histones instead. Proteins mostly studied for roles in mRNA maturation, in particular RALY, interacted endogenously with NORS and knockdown of RALY abolished the effects of NORS on steroid biosynthesis program. Finally, the authors interrogate GWAS and QTLs datasets. In the case of NORS, there are five SNPs and they are among the top SNPs associated with primary or secondary steroid-related phenotypes including puberty timing and measurement, androgenetic alopecia, and adult stature.

Overall the manuscript attempts with partial success to characterize a lincRNA and study its function mainly in a cancer setting. The study is interesting because it identifies a modulator of the sterol biosynthesis in cancer biology, but we fear that there are some major conceptual issues due to a rather erratic plan of investigation that does not end up answering what we think are the three critical questions: What does NORS do? How is NORS regulated? What’s the impact on normal tissue development?

Answer: We would like to sincerely thank the reviewer for the overall positive opinion and detailed attention to our manuscript. The manuscript has been thoroughly revised and much

more information has been added in its new version, in a more organized manner. Particularly, the reviewer's input led us to more carefully examine the estrogen response as a major regulator of NORS. We also realized that this dual theme (O₂ and estrogen signaling) is in fact a more general feature of hypoxia-inducible lincRNAs, such as NEAT and MALAT1 (albeit these aspects are rarely if ever featured by the same study).

Overall, we aimed to provide clearer answers to the fundamental questions posed by the reviewer (underlined above):

- a) NORS is an O₂- (new Fig. 1 a-d and Fig. S1) and E₂-sensitive (new Fig. 3) transcript. Further evidence for the role of HIF2 (in contrast to HIF1) is herein included (new Fig. 2a-d and Fig. S4 a-b) These results are described in "Identification and Characterization of NORS" and "Upstream molecular determinants of NORS abundance" under Results section (lines 103-200).
- b) NORS functions as a negative feedback regulator of sterol/steroid biosynthesis (Please see "NORS Coordinately Modulates the Sterol/Steroid Biosynthesis Program", under Results section, lines 201-240, new Fig 4, new Fig.S6a-b, new Fig. S8 and new Fig. S9). We expanded on the previous preliminary mechanistic insights and generated additional evidence for RALY as an interface between NORS and sterol biosynthesis gene expression. (Please see "Mechanistic insights into NORS effects", line 282-302, under Results section, and new Fig. 6 and new Fig. S12).
- c) It is predominantly relevant for the biology of hormonal tissues, as evidenced by its expression pattern and rapidly expanding GWAS information (please see "NORS Biological Relevance: Lessons from Naturally Occurring Genetic Variations", under Results section, lines 303-339, Table 1, new Fig 7 and new Fig S13). The manuscript now includes the first functional study of naturally occurring genetic variants at this locus.

Considering that the third point has not been properly addressed and more extensive experiments need to be done in order to clarify the supposed connection between this lincRNA and the derived phenotypes (from GTEX), we recommend that the authors to consider this third point for further development (and a separate manuscript). Indeed, this reviewer thinks the overarching aim of this manuscript is too ambitious, i.e. characterize NORS at a physiologic and pathologic level.

Answer: We thank the reviewer for his/her recommendation of considering the further investigation of connections between NORS and the derived phenotypes in a separate manuscript. However, as a preliminary validation of the naturally occurring genetic variants at NORS locus, a first functional study was also included in the revised form of the manuscript (new Fig. 8) that gives a more consistent message about this locus.

Although very interesting, the connection of NORS with physiological processes is supported only by preliminary data. It might be worth to validate the analysis of available datasets done in fig 7 with experiments aimed at characterizing regulatory elements, engineering or reproducing SNPs, using even mouse models of steroid-related phenotypes like timing of menarche and genital development. In the present state, these meta-analyses are too preliminary and too correlative to support any conclusion. It is this reviewer opinion that characterization of NORS should be focused more on the cancer setting (as the totality of experiments uses almost exclusively cancer cell lines) rather than try to speculate about a possible physiological role.

Answer: As recommended by the reviewer, this aspect of the manuscript was significantly developed. The revised manuscript contains a preliminary functional study of the impact of these snps. Furthermore, we strengthened the connection between these snps and the locus, with the expert contribution of Dr. Diana Cousminer, the first to formally connect these genetic variations to puberty onset timing. Interestingly, while she identified rs246185 as the most

significant GWAS snp (subsequently replicated by each study on the subject), she also hypothesized that this may not be the “effect” snp. Our new bioinformatic and experimental studies summarized in the revised manuscript support her hypothesis regarding a stronger candidate in close proximity (please see lines 329-339, new Fig. 8 and new S13). Since our original submission, the evidence connecting this locus to human phenotypes has continued to strengthen but the dilemma regarding the relevant genetic element still lingers. A formal connection between these genetic variations and lincRNA should elicit interest in multiple fields.

This consideration brings to another important note that more characterization of the link with the estrogen response/nuclear receptor should be more explicit. Indeed, the authors’ deduction that “there seems to be a clear connection between NORS locus and estrogen/ESR1” and their knowledge that “NORS has been reported among the transcripts significantly downregulated in endocrine-therapy resistant, long-term estrogen deprived MCF7 cells. NORS also exhibits higher expression in ER/PR positive breast cancers” should be enough to further investigate this key point.

Answer: We thank the reviewer for suggesting to us an increased focus on the estrogen connection, that turned out to be a more significant theme than we originally considered. While perhaps not sufficiently appreciated, hypoxia and estrogen response are in fact rather intimately interconnected. Many if not most previously characterized hypoxia-inducible lincRNAs are also estrogen responsive (although these aspects tend not to be addressed in the same study). The hypoxic transcriptional response and estrogen response are among the most significantly overlapped. GSEA, for example. Many targets that the hypoxia researchers routinely use, CA12, VEGF, ADM, CCND1, STC1, to mention only a few, are under both HIF and ER regulation. These HREs are in close proximity to steroid hormone response elements that are also important for its regulation. The reviewer may also be excited to hear about an independent confirmation of a connection between the estrogen response and NORS genomic region, which emerged while this paper was under revision. The major ChIPseq peak near NORS locus corresponds to a trans-mega ER enhancer identified by M. Rosenfeld’s group while our manuscript was under revision (please see lines 182-188, new Fig. S6 and ref 28).

Of three major connection found (NORS and glucose metabolism, estrogen response and lipid metabolism) only the lipid metabolism has been investigated more in depth. Considering that all cells were cultured in high glucose media, whether the consensus at least for the breast cancer cell lines, is to culture them in 1 g/L glucose (normal glycemia in humans is 0.7-1.2 g/L), one cannot exclude that the real driving force of the NORS expression is high glucose itself. To exclude that glucose is a necessary and sufficient condition to activate NORS sterol biosynthesis regulation, all experiments in fig 1 should be done in low glucose conditions as well.

Answer: In the revised form, we eliminated the normal tissue correlations and metabolic responses (glucose) and concentrated on the two central themes described above.

Although the oxidative-phosphorylation pathway is one of the most upregulated in the gene set enrichment analysis in both cancer cells and normal tissues analysis (fig 3c and 3e), it is not mentioned in the manuscript nor it is investigated. Only one experiment (fig S8) was done to test the effect of Metformin (and only in the MCF7), using low and high glucose and 1% O₂ conditions. Unfortunately, there is no follow up or any mention to a possible explanation of these effects so this reviewer suggests to either expand this point or edit the figure.

Answer: We agree with the overall impression of the reviewer (point also raised by reviewer #3) that the metabolic angle (effects of glucose, lipids, metformin) was underdeveloped and not

clearly connected to the main theme of the paper. We therefore decided to remove this section for now and revisit it in detail in future studies.

Furthermore, the reviewer thinks that more consistency should be used for the oxygen deprivation conditions. The variability between the experiments and the cell lines used ranges from 0.2% to 2% O₂. According to what stated by the authors in the text “While HIF-1 α is generally thought to mediate acute adaptation to severe hypoxia, HIF-2 α appears predominantly involved in the chronic responses to milder, often physiologically relevant, oxygen deprivation. Specifically, HIF-2 α exerts context-dependent positive or negative effects on cell growth and functions as a driver of stemness in normal or neoplastic tissues [15, 16].”, and considered their results point at HIF-2 α only regulating NORS’s hypoxic response, I suggest that a milder oxygen deprivation should be used (2%O₂) across the study.

Answer: Majority of the hypoxia experiments in this study used 1% O₂. Only the early data were generated using 0.2%. For the revised manuscript we repeated some key experiments including the effect of NORS knockdown on cholesterol synthesis genes in 2% O₂. The conditions are specified more clearly as requested (please see lines 104-121).

Regarding the connection of NORS with the lipid metabolism, the results show an upregulation of the downstream enzymes however, considered the extensive literature regarding the role of the master regulators of the sterol biosynthesis SREBPs, it would be essential to understand if NORS collaborates with SREBPs for example by seeing if these TFs localize in the nucleus under hypoxic conditions (fig 1 g). This reviewer found very interesting the relationship between NORS and RALY. To further confirm this mechanistic link, it would be nice if the authors could investigate how NORS regulates sterol biosynthesis through RALY interaction (Does RALY interact with the gene’s promoters, RALY ChIP?). Overall, this reviewer would be interested in re-considering a majorly revised manuscript. It is noted that the premised and initial findings are of interest but there is a strong need to consolidate the molecular findings without the need of major physiological extrapolations.

Answer: We agree with this point, which was significantly expanded in the revised manuscript. Our new collaborator (and coauthor) Dr. Paolo Macchi has studied RALY extensively and recently described its RNA partners in MCF7 cells, using RNA-immunoprecipitation (RIP)-Seq. Their published supplemental information lists NORS (RP11-65J21.3) as one of the significantly enriched RNAs interacting with endogenous RALY, a valuable independent validation for our data. For the revision of the manuscript our groups collaborated closely with several important outcomes. First, key results from our first manuscript (NORS induction, cholesterol gene response to NORS inhibition, etc) were independently reproduced in Italy. Second, we provide preliminary information about NORS’s ability to measurably affect the interaction between RALY and cholesterol biosynthesis transcripts at least under decreased oxygen tension (please see lines 293-302, new Fig. 6). RALY is an RNA binding protein (not detectable in direct contact with DNA to our knowledge) known to regulate mRNA biogenesis steps, thus NORS’s interplay with RALY may affect elongation and splicing (but not stability in a consistent/significant fashion).

Regarding the possibility of NORS interfering with SREBP1/2 activity, a point rightly raised by the reviewer, our proteomic screen did not yield any peptides corresponding to SREBP1 or SREBP2 (please see Table S8). Furthermore, an inspection of the transcripts upregulated by si/shNORS or downregulated by NORS overexpression (please see RNAseq tables), several canonical SREBP1/2 targets showed no significant response, including SREBP1/ 2 themselves and LDLR. We were not able to show that NORS measurably blocks SREBP1/2 binding to its canonical sites in HMGCR and HMGCS1. While these negative results were obtained with both

ChIP assays and commercial assays based on consensus binding, we did not have sufficient replicas in each case, thus in our opinion did not warrant inclusion in the manuscript. However, we will include them in supplemental information (with the above caveat) if the reviewer recommends.

We reiterate that the focus of this study was on probable direct NORS partners that are critical for its ability to regulate the sterol biosynthetic program. A more detailed dissection of NORS effects on specific steps of gene expression will be the subject of future studies.

Minor comments:

Introduction: “The outcome is a rare case, to our knowledge, of an informative convergence between the experimentally-determined function of a lincRNA and the phenotypic associations of the corresponding locus.” Overstated, as there are not conclusive data on phenotypic association (the manuscript discuss invasion but then highlights alopecia?).

Answer: The term “phenotypic associations” pertains to human phenotypes from Genome-wide association studies (GWAS), in our case mainly traits with a hormonal connection (age at menarche, genital development, age at voice break, androgenic alopecia, etc). We clarified this potential source of confusion in the revised version (lines 97-99).

Results:

Fig. 1: (e) the method used for the experiment done in patient-derived cell lines and CAFs should be consistent with the other cell lines (qPCR).

Answer: These cells were only used in the very early experiments, as the focus gradually shifted to hormonally-responsive tissues. We kept these early results going back almost 8 years simply to show that the response of NORS to oxygen deprivation can be easily evidenced even using “old fashioned” approaches. We moved this panel to supplementary data (new Fig. S1).

Legends (c-e) not correct.

Answer: We have changed the fig 1 and also revised the fig legend.

RNA scope, where are the negative and positive controls?

Answer: We added the negative probe image in the revised form of the manuscript. Both probes (negative and NORS) were commercially designed by the company. We included NORS overexpression sample as a positive control to show the designed probe is NORS-specific (new Fig. S1d).

Fig. 3b: 84 TNBC and 41 PAAD tumor samples are used, there is no mention in the methods, where they come from? Are they primary, metastatic? The paragraph title “NORS Modulates Sterol Biosynthesis” should probably be modified into something more consistent with the paragraph content. Indeed, the experiments described here are aimed at studying the effects of NORS on sterol biosynthesis in cancer and on invasion and tumour growth.

Answer: These are TCGA datasets, we added the information in the revised form of the manuscript (lines 122-128) and in the new figure S1e legend.

Methods: RNAscope, missing information (kit and products catalog number, probes, dilution and channels used)

Answer: Additional information added (lines 478-484).

WB, missing information about concentration of antibodies used

Answer: Added in Table S11.

Xenograft models, how many animals used?

Answer: 10 mice were used in each group, information added to materials and methods section (lines 517-526).

Discussion: It needs to be revised according to the comments, reconsidering all the discussion done about the puberty onset and physiology setting that covers more than half of the total space dedicated to this section. The importance given to this aspect of the paper is not justified by any of the experiments or the experimental design of the study.

Answer: The discussion was restructured and references incorporated as recommended.

References:

written twice, of which the first one of 69 is the incorrect one.

Regarding the upregulation of the sterol biosynthesis pathway promoting resistance to hormonal therapy in cancer, this reviewer suggests adding the following relevant citations:

Nguyen, V. T. et al. Differential epigenetic reprogramming in response to specific endocrine therapies promote cholesterol biosynthesis and cellular invasion. *Nature Communications* 10044 6,

Nelson ER1, Wardell SE, Jasper JS, Park S, Suchindran S, Howe MK, Carver NJ, Pillai RV, Sullivan PM, Sondhi V, Umetani M, Geradts J, McDonnell DP. *Science*. 2013 Nov 29;342(6162):1094-8. doi: 10.1126/science.1241908.

27-Hydroxycholesterol links hypercholesterolemia and breast cancer pathophysiology.

Answer: The two references were included in the manuscript.

We thank once again the reviewer for the very helpful review of our manuscript and hope the new form of the manuscript will be considered more organized and focused.

Reviewer #2

The manuscript entitled “NORS: a LincRNA Regulator of Sterol Biosynthesis” by Wu et al is announced as an original work contribution. In this work the authors identify a long-coding RNA which they name NORS by screening for dynamically regulated transcripts in response to oxygen tension in a human cancer cell line. Using co-expression network analysis from publically available datasets and lncRNA perturbations studies mostly done in vitro the authors assert that NORS plays a role in sterol biosynthesis.

Overall the work is novel, timely and provides interesting insights on the contributions of lncRNAs in human metabolic regulatory networks. This is particularly noteworthy since extrapolation of murine effects to human disease is often a challenge in the field of lncRNA biology owing to their poor conservation. Nevertheless, the manuscript falls short on multiple fronts including a number of technical and methodologic concerns as well as lack of detailed mechanistic insight.

Answer: We sincerely thank reviewer 2 for the overall appreciation of our study (“novel, timely”) and took his/her valuable recommendations/critiques into account to improve the manuscript.

1. The work almost exclusively uses RNAi for functional studies. siRNAs and shRNA require machinery and are known to have substantial off target effects that notoriously influences sterol regulatory pathways. In addition, their sole use is not considered

acceptable for thorough interrogation of lincRNA functions. If the authors wish to make a claim about regulation of sterol biosynthesis by NORS, genetic inactivation tools such as CRISPR are absolutely required and can be readily accomplished nowadays. What is the effect of knockout of NORS on sterol biosynthetic genes and does unbiased gene expression in this context reveal other pathways? Does overexpression of NORS in this context rescue the phenotype?

Answer: With regards to the need for CRISPR. CRISPR is very feasible when investigating coding loci, but much less so when studying a noncoding locus with multiple components (lincRNA, miRNAs). There is no effect of small indels, that disrupt ORFs, deletion of critical exons, or regulatory regions. For lincRNA loss of function studies people often resorted to CRISPR after failed attempts with siRNA/shRNA/GAPMers. While CRISPR is a potentially powerful tool, the approach should not automatically be considered superior to all siRNA/shRNA-based strategies. Many high-quality studies have been published on lincRNA without involving CRISPR interventions. Just one of the many examples, a recent Nature Metabolism paper on lincRNA CHROME a regulator of cholesterol pathways (with an accompanying editorial by PT). The study relied on well-controlled siRNAs and Gampers, similar to our approach. Our strategy based on shRNA/siRNA is tailored for studying the consequences of quantitative variations in NORS abundance (rather than complete obliteration of the locus which is less likely to occur naturally). It also avoids interference with intronic miRNAs (two miRNAs in this case). Furthermore, a genomic intervention in this case may also lead to a potential disruption of chromatin interactions involving the miR193BHG locus. A note of caution regarding the superiority of CRISPR postulated by the reviewer: It is becoming increasingly clear that under some circumstances such genomic interventions can disturb genomic neighborhoods of genes (topologically associated domains) and that CRISPR is by no means the perfect tool as perhaps originally thought (PMC6884486; PMC6731277; PMC6907074).

With respect to the biological relevance of a mouse KO, NORS is well-conserved among primates, but less so between humans and more distant mammals including mice. Even less conserved is LINC02130 locus, the putative 3' enhancer hosting the strongest regional HIF and ESR signals.

Regarding a role for NORS in the liver, we did not make an argument for NORS as relevant for liver cholesterol synthesis, but for cellular sterol/steroid biosynthesis the extrahepatic facet of this pathway, that plays particularly important roles in hormonally responsive tissues. This message is collectively sent by population genetics. Virtually all GWAS published for puberty-related phenotypes, as well as other hormonal phenotypes identify this locus as highly significant (in some case among the very top). In contrast, none of the many GWAS on cholesterol/lDL, hDL, statin- related phenotypes yielded significant hits at this locus.

Regarding the specificity of our loss of function studies. The reviewer argues that these are pathways generally affected by siRNA. First, we have not seen many cases of lincRNAs that have such a coordinated effect on cholesterol metab pathway, with compatible phenotypic associations of the corresponding locus. Second, NORS siRNA was compared with another siRNA (neutral) and, siRNA strong versus weak. Third, the effects are largely recapitulated using a nonoverlapping shRNA versus a neutral shRNA. Fourth, siRNA and OE elicit generally opposite effects. Lastly, the following also indicate that the effect is specific: it is largely eliminated by depleting the cells of RALY. It is generally more robust at lower oxygen concentrations, where NORS is more abundant. All these argue and against a nonspecific siRNA effect.

Regarding the concern about RNAscope dots number as a reflection of low abundance. We do not believe that the number of dots revealed by our RNAscope is directly informative of absolute abundance. A significant fraction of NORS is chromatin-associated and potentially less accessible to the probes. Furthermore, visualization was performed using a conventional

fluorescence microscope, thus the signal(dots) does not show individual molecules but includes specific chromatin loci where more NORS molecules are likely to be found, suggesting a greater abundance of NORS molecules (compared to that of fluorescence dots). However, this assay was only intended to illustrate NORS features such as scattered nuclear distribution and relative induction by low oxygen.

2. Although the manuscript focuses on a human specific lncRNA which can be difficult to perturb, very little evidence is provided to substantiate proposed mechanisms as operational in vivo. This is particularly important since cellular models have significant limitations in studying sterol regulation and the phenotype reported here is very subtle. Since the authors are invoking a trans mechanisms of action what are the consequences of NORS overexpression in mouse liver (using adeno or AAV)? Do the authors observe changes in liver and serum cholesterol levels as well as gene expression of SREBP and target genes? In addition, the manuscript would be significantly bolstered if the authors can use the same approach in a humanized mouse liver model. These studies will provide important information on the functional conservation of this lncRNA which can be preserved despite sequence evolution as well as help tie the main conclusions of the paper with a more relevant in vivo model.

Answer: We emphasize that all the evidence points to importance for the biology of hormonal tissues rather the liver. GWAS sends a strong message with this respect. We apologize if this idea was not conveyed with enough clarity in the previous version.

3. The evidence that the lncRNA and microRNAs are independently regulated is not sufficient and based mostly on qRT-PCR data in figure 1. Are the authors proposing that the expression of the microRNA does not require the production of the host gene lncRNA? Figure 2A does not support that notion since it shows that the induction in newly synthesized RNA with oxygen tension includes the intronic regions where the microRNAs are embedded. The authors should disrupt microRNA production using CRISPR and show that effects of NORS perturbation are still preserved.

Answer: The miRNA region is in fact overlapping with regulatory features, and this region is also involved in chromatin interactions. A more basic question: what happens when the lincRNA is substantially but not completely depleted using a method that it does not affect the miRNAs. This is especially important given the enhanced evidence that changes in abundance are very likely to have human phenotypic relevance.

4. Very little information is provided on the characterization of the NORS transcript itself. What is the copy number of NORS in MCF-7 cells? The single molecule FISH supports a very low copy number. In addition, the authors should show evidence of in vitro protein transcription/translation using a positive control.

Answer: While we did not test NORS abundance using the method recommended by the reviewer, we are presenting unbiased information that is at least as relevant in our opinion. As shown in new Fig S3, according to the latest GTEX datasets, the expression of NORS is similar or higher than "classic" lncRNAs such as HOTAIR, HIF1A-AS2, CCAT1, to mention but a few. Ditto for MCF7 cell line according to LincATLAS (data not included, but easy to verify).

With regards to the concern/ query about potential peptides generated by NORS: We added analyses using several algorithms including PRIDE reprocessing 2.0, Lee translation initiation sites, PhyloCSF, and Bazzini small ORFs (please see new Fig S2c). All the above-mentioned programs concur that NORS is a noncoding transcript. We further queried public small ORFs datasets for e peptides detectable by mass spectrometry corresponding to NORS and found none. Thus, even if a peptide may be produced by NORS under some circumstances, it is unlikely to be in biologically meaningful amount.

5. The figures are very poorly rendered in particular the genome browser shots make it impossible to decipher critical information about the relationship of the lncRNA transcript and embedded microRNAs and neighboring genes. The figures need to be of much higher resolution with font size and illustrations that make it very clear where the lncRNAs transcripts are produced and their relationship to other transcripts.

Answer: The revised manuscript has high resolution figures.

6. The authors should present unbiased analysis of the BrU-seq in the supplement.

Answer: We included the complete analysis report as supplemental table (please see Supplementary Table S3), as requested. The raw data will become publicly available after the manuscript is accepted for publication. We also included BruSeq track views of MIR210HG and HIF1A-AS2, quintessential hypoxia-inducible noncoding RNA loci.

7. Please describe the source data in the methods and more detailed methodology including how gene expression is normalized and detailed primer sequences. Please be sure to report error bars for all graphs for example are the results of the first panel in 2C significant?

Answer: For all the data from previously published work, we included the GEO number either in the text or material and method section. Primer sequences are detailed in the Supplemental tables (please see Table S10). We are in the middle of organizing our own sequencing data to be uploaded and submitted to GEO and will provide the information once this manuscript is accepted. We have the description of error bar and that statistics used for each figure in the figure legend. When $p < 0.05$, it was marked by asterisk. In Fig 2b, shHIF2A effect on NORS level is only significant in MDA-MB-231 cell line, not in MCF7.

We thank the reviewer for the review of our manuscript and hope we successfully addressed all him/her queries.

.....

Reviewer #3

Xue Wu et al. provide a profound functional analysis of the lncRNA NORS. The authors show that NORS is hypoxia inducible, mediated by HIF-2 and FOSL2 transcription factors. By numbers of experimental and bioinformatics data, they characterized NORS genomic location and environment as well as NORS regulation in hypoxia and in diverse cancer types, including patient-derived carcinoma cells. In vivo, NORS correlation analysis from tumor and normal tissues data revealed that NORS contributes to a hypoxic/ tumorous network. By gene set enrichment analysis (GSEA) of NORS-correlated genes, the authors identified pathways correlated with NORS beyond the hypoxic response. NORS involvement in estrogen response and especially the cholesterol biosynthesis pathway has been further investigated in more detail. It is demonstrated that NORS interaction with RALY is crucial for its regulatory role in sterol synthesis. These findings are linked to NORS ability to inhibit cell invasion and metastasis of breast cancer cells.

The results are novel and of outstanding interest to others in the community and in the wider field. The authors characterized lncRNA NORS in depth, including up-stream factors and down-stream pathways. For instance, the data provide evidence for a new layer in the complexity of the hypoxia/ sterol biosynthesis network up to the point of NORS being a new tumor suppressor. The methods used are appropriate, state of the art and well described. The claims have been adequately discussed. The data are convincing

and I do not see the need for further experiments. In contrast, at some points I suggest to remove data to improve clarity, as outlined below. Some minor issues should be addressed.

Answer: We thank the reviewer for the interest and appreciation.

Summary: The summary does not reflect the magnificent findings of the study.

Answer: The summary was extensively revised.

Results section, "Identification and Characterization of NORS": The usage of different oxygen levels (0.02 %, 1 %, 2 %) appears somewhat confusing. The statement "...was also detected at 2% O₂, a more physiologically relevant oxygen concentration" is not correct. We all know that oxygen tension is highly diverse in different tissues, even under physiological conditions. I suggest removing Figure 1f, or, alternatively, showing an oxygen response kinetic (0.02 up to 2 % O₂) of NORS levels for cell lines MCF-7 and MDA-MB-231 as Figure 1f. The latter option would well fit to the Discussion section.

Answer: We have introduced an oxygen response kinetic of NORS levels (see new Fig.1c), based on the reviewer's comment.

Please check the link to Table S2 in "...confirmatory evidence for NORS's oxygen sensitivity (Table S2)." Do you mean Table S1?

Answer: Table S2 showed alternative names for NORS found in previous publications, which reflect its oxygen sensitivity (HypERInc or HS.190748). However, we reorganized all data and removed this table from the revised form of the manuscript.

Figure 1: Please indicate time of hypoxia in legend of Figure 1b. The attribution of Figure 1c-e in the legend is wrong ("b" is c, "c" is d, "d" is e). Please indicate that the same cycle number was used in (e).

Answer: Addressed.

Figure S1: Please provide higher resolution pictures for S1a-d.

Answer: Done

Results section, "Hypoxic Regulation of NORS":

Please improve resolution of Figure 2a, e.

Answer: Done.

In Figure 2b it is shown that NORS is unaffected by DMOG in MCF-7 cells. A 2-OG analog that inhibits prolyl-hydroxylases will activate all HIF isoforms, including HIF-2. In the same cell line, NORS is elevated by 1 % oxygen. Western blots in Figure 2c suggest that HIF-2alpha is present in MCF-7 cells and, thus, should influence NORS level similarly as in MDA-MB-231 cells. Does DMOG not induce HIF-2alpha in MCF-7 cells? Showing Western blots for HIF-2alpha in DMOG treated cells (Figure 2b) and comparing HIF-2alpha levels in MCF-7 and MDA-MB-231 cells is recommended.

Answer: This is a very sharp observation by the reviewer and certainly warrants attention. We did these experiments many times in more cells than we showed in the manuscript. The message is the same, overall DMOG (or DFO for that matter) either induce NORS modestly or not at all, in clear contrast to the dramatic effect on miR210, the prototypical hypoxia noncoding RNA. We tested the effects of DMOG on HIF1 and 2 response on both sides of the Atlantic and the simple scenario that HIF2 induction is absent does not materialize. HIF2 is induced by DMOG in MCF7 cells, as shown in the new figure. This was one of the several signs (including the ineffectiveness of HIF1) that prompted us to explore the multifaceted nature of this locus

(rather than treat it as a “miR210-like” responder and not search beyond the boundaries of oxygen-related signals). As we emphasized in the manuscript, the HIF2 connection is also apparent in the supplemental data published by Qing Zhang's team in Science not too long ago (PMC6154478), as they also found miR193BHG (our NORS) among HIF2 targets (but did not pursue it).

Figure 2c: In MDA-MB-231 cells the LaminB1 signal is strongly overexposed, leading to a burn out of chemiluminescence substrate that in turn causes white bands within the black signals. Such overexposed signals are insufficient for normalization. I suggest to provide LaminB1 signals of lower intensity.

Answer: Addressed as recommended (new Fig. 2b)

Figure 2d: Please provide significance levels for the comparison of 21 % vs. 0.2 % O₂ in VHL+ and VHL- cells.

Answer: Addressed. (new Fig. 2c)

Results section, “NORS Regulatory Signals In Vivo: Insights from Normal and Diseased Tissues”:

Authors should consider to rephrase the sentence: “...we observed that the response of NORS to decreased oxygen concentration is restricted by glucose availability, supporting a connection between NORS and glucose metabolism (Figure 3d).”. Keeping in mind that during hypoxia, glucose represents the only substrate in energy metabolism (ATP production by glycolysis only), glucose deprivation in hypoxia may cause a drop in general ATP availability that also affects overall transcription rate. Furthermore, a correlation of NORS with glycolytic factors as shown in Figure 3c is expected, as all are hypoxia inducible. For me, these data do not satisfactory show a direct connection between NORS and glucose metabolism as suggested by the authors. However, this is not the focus of this study; thus, authors may consider investigating the NORS-glucose metabolism axis separately.

Answer: Agree. Per reviewer 1 and 3 suggestion we removed the metabolic axis.

Results section, “Mechanistic insights into NORS action on the steroid biosynthetic program”:

Please provide a short description regarding TRIM28. Is the binding of NORS to TRIM28 at 1 % oxygen significant (Figure 6c)?

Answer: As the present manuscript developed the RALY connection, TRIM28 was deemphasized.

Following the authors argumentation, significance levels for si_CTRL vs. siRALY in cells treated with si_NORS_s during hypoxia and normoxia should be indicated (dark green columns in Figure 6f). For clarity, I suggest not to show data for si_NORS_w in Figure 6f, as it does not contribute to the conclusion.

Answer: Agree, we removed the si_NORS_w data from this figure (please see new Fig 6d)

Significance levels are also missing in Figure 6b.

Answer: Added as suggested (please see new Fig. S12a)

Discussion, para 4: I suggest to rephrase “physiological hypoxia”, as hypoxia represents a condition when oxygen demand exceeds supply. “physiological low pO₂” would be more appropriate.

Answer: We incorporated the concept, and we appreciate the suggestion. We concur with the reviewer that this term characterizes accurately naturally encountered oxygen tension and variations thereof.

Methods, “Subcellular fractionation”: Please mention the conditions of centrifugation to obtain the cytoplasmic fraction (10,000 x g?).

Answer: Inserted as requested (line 466)

The link to Table S2 for the list of gene-specific primers is wrong. Supplementary Table “oligos” would fit.

Michael Föhling

Answer: Corrected as requested.

We would like to sincerely thank again the reviewer for the overall positive opinion to our manuscript.

Reviewer #4

The authors report the first characterization of a noncoding RNA named NORS. They showed that NORS-regulation involved growth signaling and that NORS repressed steroid biosynthesis (including sterol) transcriptional programs. They used genome-wide association studies to support the biological relevance of NORS with the identification SNPs on enzymes form the sterol pathway overlapping NORS’s regulatory region. Finally, they report the identification of an interaction of NORS with the RALY protein, which was shown to repress cholesterol biosynthetic genes expression in mouse liver.

Although this study opens new perspectives in the identification of possible new actors involved in the regulation of steroid/sterol metabolism, it is mainly correlative and thus preliminary. Importantly, this study lacks experimental evidence that NORS controls cholesterol neo-synthesis.

Answer: We thank reviewer #4 for the interest and overall appreciation of our study and took his/her valuable recommendations into account to improve the manuscript. We show additional evidence that NORS predominantly affects the cholesterol biogenesis program at transcriptional rate level and that NORS is most likely relevant for the extrahepatic “facets” of sterol/steroid biology rather than liver cholesterol biosynthesis, and plays particularly important roles in hormonally responsive tissues.

Reviewer #5

I think Wu et al make an interesting observation, which is the regulation of NORS levels under different conditions. Unfortunately, the quality (resolution) of many of the figures is pretty low and it is often difficult to read/evaluate. My review will focus mostly on the proteomic analysis of NORS; however, I would like to highlight that authors should improve the quality of the figures by using images with higher resolution.

Answer: We take this opportunity to thank the reviewer for helping us improve the manuscript on many levels, even if his/her stated primary expertise is proteomics. First, we increased the

resolution for most of our figures, we apologize for the sub-par graphics in the first submission.

PROTEOMIC ANALYSIS OF NORS

I find some limitations in the NORS ‘interactome analysis’ that require discussion/clarification or/and experimental validation. It is not the same to identify proteins that interact with NORS and to establish the NORS interactome. The first refers to cherry pick proteins from the proteomics through a hypothesis driven approach aiming at finding NORS regulators and understanding that the dataset is corrupted by false positives. The second refers to the comprehensive determination of the repertoire of proteins that interact with NORS, minimizing false positives to provide a resource for the community. Each approach have different requirements. While the first would focus on the downstream experimental validation of the protein’s function (which I don’t think is enough here), the second will require extensive validation of the capture as well as stringent analysis of the data. As it is, I don’t think this work fulfill the requirements to call for a ‘NORS interactome’.

Answer: We thank the reviewer for this detailed and valuable advice, we recognize that the term “NORS interactome” was misused and thus eliminated from the revised manuscript.

1. First, NORS is expressed by transitory transfection, which means that NORS levels are expected to be non-physiological. Experiments determining endogenous and ectopically expressed NORS and how they compare are required to balance the conclusions.

Answer: Most of the experiments were done in the context of endogenous NORS levels, including the siRNAs, shRNAs. Furthermore, we are presenting multiple lines of evidence for an endogenous interaction between RALY and NORS.

In the revised manuscript, the subject of multiple queries has been eliminated or de-emphasized thus we are providing a collective answer to these. Critically important, we generated interaction evidence from the opposite direction, based on recently identified RALY interacting RNAs in the same cell type (MCF7).

2. As the capture relies on MS2, native purification conditions are used. This implies that authors purify protein that bind directly and indirectly to NORS and this should be stated in the text.

3. It is known that MS2 capture when using in cellulo approaches are very noisy due to the purification of indirect binders and unwanted RNAs. To convince me about this proteomic analysis, authors should exhaustively analyze the RNA pool isolated by MS2 capture in their experimental conditions. This can be done through RNAseq (optimally) or bioanalyser (at least to show that the dominant RNA population has the size of NORS). Enrichment over ribosomal RNA only tells me the degree of enrichment against that particular (highly abundant) RNA but does not provide any information about the specificity of the capture beyond that. Just looking at the numbers, knowing the abundance of rRNA and NORS (FISH analyses) per cell, rRNA would still be dominant in the pull down in spite of the enrichment showed in Figure 6b (30-fold). This would explain why authors identify basically the whole ribosome in the pull down. Would this mean that the lincRNA is translated or just relate to contaminant RNA? As a positive aspect, I understand from the figure (not clear in the figure legend) that the control is absence of an RNA with MS2. This is good as the contaminant RNA will mostly come from MS2-bp unspecific binding when present in excess. I suggest to the authors to clarify this point and describe in more detail the experimental approach and the controls used.

As the reviewer noted, the control is a pulldown from cells expressing an artificial RNA (24x repeat), thus the artifacts are most likely the result of overexpression rather than nonspecific

binding of RNA binding proteins. Furthermore, the entire strategy for identifying protein interactors has been reframed, with the MS2 – based pulldown representing only one component of the search. Again, we fully agree with the reviewer regarding the limitations of this overexpression pulldown.

4. To convince me/us of the high stringency of their method, they should show the proteins purified by the MS2 capture in experimental and controls by a general protein staining, in particular high sensitivity silver staining. They should detect an enrichment of specific bands in the specific capture over the controls. Figure 6b shows that enrichment of candidate RBPs in the experimental over control is residual (see RALY and TRIM28).

5. This is unconvincing to me as I would expect a black (negative control) and white (experimental) situation if the purification has enough stringency. Can authors comment on this?

TRIM28 was de-emphasized, as we concentrated in more detail on the more probable direct interactor, RALY, for which additional pieces of evidence were added in the revised manuscript.

6. SAINT employs protein length to normalize/correct the spectral counts and this affects probability, which is good as the bigger of protein is, more chances of getting spectral counts from different peptides are. However, authors employ here a very loose SAINT cut off (0.5). Choi et al., Nat Meth 2011, defined 0.8-0.9 SAINT values as stringent cut offs equivalent to 5% and 2% FDR, respectively. From my point of view, a cut off of 0.5 is just not acceptable in a noisy experiment such as specific RNA capture by MS2. Note that most of the proteins in the 0.5 statistical threshold display very small fold change over controls (that is supposed to lack the RNA of interest). Under the limitations of specific RNA capture (low signal over noise, contamination with unwanted RNAs, use of native conditions, etc), one should apply stringent statistical standards to minimize false positives.

7. One of the problems is that RALY, the proteins that authors focus afterwards on, has 0.5 SAINT value (2.39- and 1.63-fold change; which seems low to me when comparing to a negative control). Based on this value, there is not strong support from the proteomic data on the NORS/RALY interaction. This raises questions about why authors selected this protein. This needs clarification as for me, other proteins in the list would represent much better candidates for follow up studies (e.g. TRIM28).

We certainly agree with the weakness of interactor candidates selected solely from overexpression systems. As stated above, TRIM28 is not pursued in detail in the revised manuscript, while the case for RALY as critical NORS interactor is expanded. Regarding the modest SAINT score for RALY, we noted its absence in the crosslinked pulldowns in sharp contrast to its strong presence in pulldowns without crosslinking. A potential explanation is that RALY is a very basic protein, thus amenable to over-crosslinking. In hindsight we think it was not rigorous to generate a SAINT score mixing data from two different conditions.

Overall, the section on NORS protein partners was significantly restructured and the choice of RALY is now justified by a combination of experimental and computational criteria.

8. If authors want to say that the data provided is the ‘NORS interactome’, they should validate multiple proteins falling under different SAINT scores and exhibiting different fold changes by an orthogonal approach.

We thank the reviewer for this suggestion. We agree with his/her reservations for the term “NORS interactome” and we have eliminated it from the revised version.

9. Most of the isolated proteins appear to me as highly abundant (authors mentioned histones, but they also capture the near-complete ribosome). A density plot comparing protein or RNA abundance of the whole proteome and NORS proteome would help to rule out whether this approach just enriches for abundant proteins. This can be done with data already available in repositories. Moreover, as NORS is a (mostly) nuclear RNA, nuclear GO annotation should be prevalent over cytoplasmic in the 'NORS' interactome.

Regarding the high abundance of ribosomal proteins, we agree with the reviewer that this aspect needs to be clarified. The outcome of the pull-down represents a collection of in vivo partners, and overexpression artifacts. The high abundance of ribosomal components is probably due to the artifactual presence of transduced NORS outside the nucleus. This emphasizes the importance of multiple criteria for the choice of biologically relevant interactions: known RBP status, as this class includes the most probable direct NORS partners. This group was further narrowed down to the pool computationally predicted to bind NORS sequences. When all these are considered, RALY emerges as one of top candidates, as shown in the restructured Supplementary Table.

10. The validation by RIP is also unconvincing for me. RALY is a pretty good RBP in terms of UV crosslink-ability and I thus expect RALY produce quite extensive fold enrichment in RIP assays. However, authors only see 3-4-fold enrichment in normoxia and 7.5 in hypoxia – which correlate with higher NORS levels. RIP fold enrichment for TRIM28 is not convincing at all in spite of the high SAINT score. Said that, I think this study would gain interest if authors would strength their point by elucidating whether RALY/TRIM28 binds NORS directly, and where these interactions occur. I suggest authors check first whether there is any RALY or TRIM28 iCLIP, HITS-CLIP, eCLIP dataset available in ENCODE to take advantage of readily available data. If not, performing this experiment with add value to the paper.

This was particularly valuable suggestion and we thank the reviewer for the insight. As mentioned above, the reality of NORS-RALY interaction is indeed supported independently by published protein-RNA interaction datasets. Our new collaborator Paolo Macchi identified NORS (listed as RP11-65J21.3) among the significantly enriched RALY-binding transcripts and his team provided independent experimental confirmation for this partnership.

OTHER COMMENTS:

11. Figure 1. Is abundance of the RNAs located around NORS increased upon hypoxia?

Answer: As shown on RNAseq tracks and Bru-seq & RNAseq supplemental info, PARN and MKL2/MRTFB do not appear induced by hypoxia. Consistently, these 2 genes are rarely if ever listed among hypoxia-inducible genes in the literature. A similar case is represented by PHRF1 and RASSF7, the coding neighbors of miR210HG. While miR210HG is among the top recognized hypoxia-responsive loci (confirmed by our Bru-seq and RNA-seq datasets), its flanking neighbors are not.

12. Figure S1C and D are not readable. So, I cannot make any conclusion about them.

Answer: Addressed with high-resolution figures.

13. Figure 1h. To show reproducibility, it would help to provide a quantification of the RNAs dots detected in nucleus vs cytoplasm and in high and low oxygen from multiple biological replicates.

Answer: It was not our intention to use RNA scope imaging for absolute quantitation. While the signal is most likely specific, we cannot state anything about what percentage is being captured.

14. Figure 2c. Could you analyse non-responsive RNAs and cell death markers to rule out

that HIF2alpha depletion affects the capacity of the cell to deal with hypoxia and the effects are thus indirect? Same applies to FOSL2 (panel f).

Answer: At 1% we do not see significant death in these cell lines if low oxygen is not compounded by nutritional deprivation. The cells are functional enough to mount a HIF-1 response, and control proteins appear largely unchanged.

15. Page 6. 'NORS was supported by Coding Potential Assessment Tool (CPAT) analysis which yielded a very low coding potential score, comparable to those of MALAT1 and NEAT1 (Figure S1e).' Can you support this using any available ribosome profiling data?

Answer: As replied to review # 2 above, the collective information extracted from sORFs datasets does not provide any evidence of peptides generated from this locus. We also added alternative bioinformatics prediction analyses for the coding potential of NORS, which returned results similar to canonical lincRNAs (new Fig. S2c, lines 946-948).

We thank once again the reviewer for the very detailed and helpful examination of our manuscript.

Reviewers' comments, second round:

Reviewer #1 (Remarks to the Author):

The authors have done a good job in addressing the vast majority of my comments. Most importantly, the manuscript now is easier to follow (for the reader). I would therefore recommend publication (recommendation limited to my expertise)

Reviewer #2 (Remarks to the Author):

Although the reviewer appreciates some aspects of the manuscript the revision is highly disappointing and does not address the glaring issues previously identified. The major claim of the paper is identifying a link between a noncoding RNA and cholesterol biosynthesis therefore providing strong evidence backing up that claim would naturally be a key point. Please look at Figure 4 the main panel showing loss of function siRNA studies. The changes in cholesterol biosynthetic genes is subtle to nonexistent. In fact, the labelled "weak" siRNA does not appear to statistically influence most of the qPCR gene panel. Asking for additional evidence reinforcing the main findings of the paper is therefore crucial and was in my original request #1. The data presented from the siRNA and shRNAs is not convincing and the use of this approach as a sole perturbation strategy may perhaps be justified in the presence of convincing phenotypic changes, rigorous study design and clear mechanistic insight but this is not the case here. I am in agreement that one needs to complement genetic perturbations with other loss of function strategies for the study of lncRNAs but insinuating that genetic inactivation or CRISPRi is an inferior approach to lncRNA interrogation would not be supported by most of the landmark lncRNA functional studies. The other major concern, it is unclear what is the significance of finding a lncRNA that changes cholesterol mRNAs ONLY subtly and ONLY in cancer cells and ONLY under hypoxic conditions. The in vivo interrogation data is weak and even if endogenous NORS may not be important in liver, all the other key ingredients claimed to be important by the authors (RALY and cholesterol genes) are in liver therefore testing a gain of function in vivo approach (can readily be accomplished using adenovirus or AAV) would show that some of the proposed in vitro mechanisms are operational in vivo which was my request #2.

Finally, the title, abstract, and text need to match the results of the paper. Suggest revising to "...: A lncRNA regulator of cholesterol biosynthesis in cancer cells under hypoxic conditions." A PubMed search for NORS yields over 1500 articles. It is a disservice to your own work and the field to call the lncRNA NORS and therefore strongly consider revising the name.

Reviewer #3 (Remarks to the Author):

The points raised in my previous round of review have been satisfactorily addressed.

Reviewer #4 (Remarks to the Author):

The authors did not address my concerns. In the absence of measurement of cholesterol neosynthesis the significance of the paper is questionable.

Reviewer #5 (Remarks to the Author):

Wu et al., have performed a substantial amount of experiments to address the comments of the five referees. I think the work is more solid and balanced now. I am glad to see that authors have applied a RIP-seq experiment to dissect the network of transcripts bound by RALY and it is very encouraging to see NORS between the top hits. It is a pity authors didn't apply eCLIP/iCLIP to obtain binding site resolution, as this could potentially provide additional insights into RALY regulation of NORS. Said that, this can be done in following up work and I understand this is a

time consuming experiment that is not essential here.

I acknowledge that authors have toned down all the section relative to the proteomic analysis of NORS. Since they neither sequenced their pull down to determine the composition of the isolates nor perform a proteomic analysis with the highest standards, I encourage that they state in the text the purpose of the experiment more clearly. In other words, to identify candidates with potential regulatory roles of NORS function and not to generate a comprehensive NORS interactome. In this context, the analysis is good enough as authors provide substantial follow up experiments supporting the implications of RALY/NORS interaction. However, I recommend that authors explicitly state that this dataset, as a whole, is not validated and that due to the experimental caveats of MS2 purification, may contain false positives. Moreover, authors should encourage readers to validate the interaction of other candidates with NORS (as they did with RALY) before engaging in time consuming functional assays.

Lane 292 'lncRNAs with affinity for RALY'. I don't think this sentence is correct. Authors are mixing here specificity and affinity, which are two different things. I suggest they just indicate that 'NORS is recognised or bound by Raly in cells'.

We take this opportunity to thank our reviewers for their valuable input and predominantly positive opinion about our study. The present revision includes conceptual clarifications and a panel of additional metabolite data, in response to lingering concerns from reviewers 2 and 4, as detailed below:

Reviewer #1

The authors have done a good job in addressing the vast majority of my comments. Most importantly, the manuscript now is easier to follow (for the reader). I would therefore recommend publication (recommendation limited to my expertise).

A: We appreciate the reviewer's input and constructive critiques, as these were crucial for the new version of the manuscript.

.....

Reviewer #2

Although the reviewer appreciates some aspects of the manuscript the revision is highly disappointing and does not address the glaring issues previously identified. The major claim of the paper is identifying a link between a noncoding RNA and cholesterol biosynthesis therefore providing strong evidence backing up that claim would naturally be a key point. Please look at Figure 4 the main panel showing loss of function siRNA studies. The changes in cholesterol biosynthetic genes is subtle to nonexistent. In fact, the labelled "weak" siRNA does not appear to statistically influence most of the qPCR gene panel. Asking for additional evidence reinforcing the main findings of the paper is therefore crucial and was in my original request#1. The data presented from the siRNA and shRNAs is not convincing and the use of this approach as a sole perturbation strategy may perhaps be justified in the presence of convincing phenotypic changes, rigorous study design and clear mechanistic insight but this is not the case here. I am in agreement that one needs to complement genetic perturbations with other loss of function strategies for the study of lncRNAs but insinuating that genetic inactivation or CRISPRi is an inferior approach to lncRNA interrogation would not be supported by most of the landmark lncRNA functional studies.

A: We thank this evaluator for his/her feedback and hope that the revision addresses his/her concerns at least in part. We certainly agree that the effects of siRNA-weak do not pass significance levels in qPCR assays but are generally in the same direction with siRNA strong. More importantly, the effects of siRNA strong, are similar with the nonoverlapping shRNA and generally opposite to NORS overexpression. As stressed previously, there are additional specificity features, including stronger effect at lower oxygen concentrations (where NORS is more abundant) and loss of effects in absence of RALY.

Furthermore, we apologize if our response was interpreted as "**genetic inactivation or CRISPRi is an inferior approach**". We simply wanted to state that no gene inactivation approach is perfect on its own and may also reveal different facets of a specific locus. In this particular case we took into account genomic features of the locus, and we were satisfied with the information collected with our approaches, similar to papers: Ali et al., Nat Commun. 2018 (PMC5830406) and Hennessy et al., Nat Metab. 2019 (PMC6691505). Moreover, in our case CRISPRi would have downregulated or shut down completely the lincRNA and the 2 miRNAs from this locus.

The other major concern, it is unclear what is the significance of finding a lincRNA that changes cholesterol mRNAs ONLY subtly and ONLY in cancer cells and ONLY under hypoxic conditions. The in vivo interrogation data is weak and even if endogenous NORS may not be important in liver, all the other key ingredients claimed to be important by the authors (RALY and cholesterol genes) are in liver therefore testing a gain of function in vivo approach (can readily be accomplished using adenovirus or AAV) would show that some of the proposed in vitro mechanisms are operational in vivo which was my request #2.

A: We need to reiterate key points from the previous replies. First, NORS locus exhibits limited conservation between human and mouse genome. Second, NORS is most likely relevant for the biology of hormone-responsive tissues (breast, gonads, prostate etc), rather than liver. Therefore, a liver-specific mouse model is unlikely relevant for studying the human locus.

Finally, the title, abstract, and text need to match the results of the paper. Suggest revising to "...: A lincRNA regulator of cholesterol biosynthesis in cancer cells under hypoxic conditions." A pubmed search for NORS yields over 1500 articles. It is a disservice to your own work and the field to call the lincRNA NORS and therefore strongly consider revising the name.

A: We thank the reviewer for recommending us to revise the lincRNA name, we appreciate his/her suggestions. First, we agree with the ambiguity of "NORS", we have changed it *lincNORS* which returns zero hits on Google and Pubmed. Furthermore, based on the feedback from this reviewer, as well as reviewer 4, we realize that the term "sterol biosynthesis" may convey the wrong message for many readers while failing to deliver the main one. Thus, in the revised version of the manuscript NORS stands for "**Noncoding O₂-Sensitive Regulator of Sterol Homeostasis**", which should preempt any confusion with the "traditional" liver cholesterol biosynthesis and leaves the door open for a potential impact on other branches (suggested by the response of STARD4, STARD3NL). This title should also reflect upstream and downstream connections between lincNORS and steroid biology, as described in our first revision. Some evidence was shown in noncancer cells, and biochemical aspects are for practical reasons dissected in cancer cells, GWAS indicate the role in physiology.

Reviewer #3

The points raised in my previous round of review have been satisfactorily addressed.

A: We sincerely thank the reviewer for his interest and detailed examination of our manuscript.

Reviewer #4

The authors did not address my concerns. In the absence of measurement of cholesterol neosynthesis the significance of the paper is questionable.

A: As stated previously, we reiterate that our manuscript does not attempt to portray *lincNORS* as a significant regulator of cholesterol neosynthesis in hepatocytes. We also recognize that our title may have contributed to the confusion, thus we propose a small change which should clarify the context from the outset: *lincNORS: A Noncoding Regulator of the Cellular Sterol Homeostasis*. We emphasize that this reformulation does not affect the manuscript, but helps separate this lincRNA from liver cholesterol biology. This title also reflects our findings that NORS has both upstream and downstream connections with steroid biology.

Regarding the assay suggested by the reviewer: while we did not perform flux for this study, we are now including information on additional metabolites (Fig4), obtained during cholesterol quantitation (but not shown previously). These data appear overall consistent with the transcriptomic effects of *lincNORS*. Since the lincRNA impacts multiple genes of the pathway in

the same direction it should predominantly affect end products of the pathway (cholesterol rather than lanosterol). Another metabolite of cholesterol (25OH Ch) does not change, as expected from the lack of response in gene expression of the corresponding hydroxylation enzyme CH25H from RNA-seq data (Table S5 from supplemental tables).

We acknowledge that we do not have formal proof of impact on the biosynthesis by cholesterol flux, thus we have revised the title to a more prudent and broader term sterol homeostasis. There is little doubt that the major programs fine-tuned by *lincNORS* are part of sterol homeostasis, based on gene expression signatures and metabolites quantification. We hope that the reviewer agrees with this conclusion.

Reviewer #5

Wu et al., have performed a substantial amount of experiments to address the comments of the five referees. I think the work is more solid and balanced now. I am glad to see that authors have applied a RIP-seq experiment to dissect the network of transcripts bound by RALY and it is very encouraging to see NORS between the top hits. It is a pity authors didn't apply eCLIP/iCLIP to obtain binding site resolution, as this could potentially provide additional insights into RALY regulation of NORS. Said that, this can be done in following up work and I understand this is a time-consuming experiment that is not essential here.

I acknowledge that authors have toned down all the section relative to the proteomic analysis of NORS. Since they neither sequenced their pull down to determine the composition of the isolates nor perform a proteomic analysis with the highest standards, I encourage that they state in the text the purpose of the experiment more clearly. In other words, to identify candidates with potential regulatory roles of NORS function and not to generate a comprehensive NORS interactome. In this context, the analysis is good enough as authors provide substantial follow up experiments supporting the implications of RALY/NORS interaction.

A: In the revised form of the manuscript, “NORS interactome” has been removed, being replaced with candidates.

However, I recommend that authors explicitly state that this dataset, as a whole, is not validated and that due to the experimental caveats of MS2 purification, may contain false positives. Moreover, authors should encourage readers to validate the interaction of other candidates with NORS (as they did with RALY) before engaging in time consuming functional assays.

A: The recommended statements had been added in the revised form of the manuscript (page 12, paragraph 2 and page 13, paragraph 2).

Lane 292 'lncRNAs with affinity for RALY'. I don't think this sentence is correct. Authors are mixing here specificity and affinity, which are two different things. I suggest they just indicate that 'NORS is recognised or bound by Raly in cells'.

A: At the reviewer's suggestion, the text was modified: “lincNORS is also among the top hypoxia-responsive lncRNAs bound by RALY” (page 13, paragraph 1).

We sincerely thank the reviewer for his/her helpful opinion and very detailed attention to our manuscript.

Reviewers' comments, third round:

Reviewer #4 (Remarks to the Author):

The points raised in my previous reviews have been satisfactorily addressed